



# Regional Imprints of Changes in the Atlantic Meridional Overturning Circulation in the Eddy-rich Ocean Model VIKING20X

Arne Biastoch[1,2], Franziska U. Schwarzkopf[1], Klaus Getzlaff[1], Siren Rühs[1], Torge Martin[1],
Markus Scheinert[1], Tobias Schulzki[1], Patricia Handmann[1], Rebecca Hummels[1], and Claus W. Böning[1,2]

[1]GEOMAR Helmholtz Centre for Ocean Research Kiel, Kiel, Germany
[2]Christian-Albrechts Universität zu Kiel, Kiel, Germany

**Correspondence:** Arne Biastoch (abiastoch@geomar.de)

**Abstract.** A hierarchy of global 1/4° (ORCA025) and Atlantic Ocean 1/20° nested (VIKING20X) ocean/sea-ice models is described. It is shown that the eddy-rich configurations performed in hindcasts of the past 50-60 years under CORE and JRA55-do atmospheric forcings realistically simulate the large-scale horizontal circulation, the distribution of the mesoscale, overflow and convective processes, and the representation of regional current systems in the North and South Atlantic. The

representation, and in particular the long-term temporal evolution, of the Atlantic Meridional Overturning Circulation (AMOC) strongly depends on numerical choices for the application of freshwater fluxes. The interannual variability of the AMOC instead is highly correlated among the model experiments and also with observations, including the 2010 minimum observed by RAPID at 26.5°N pointing at a dominant role of the forcing. Regional observations in western boundary current systems at 53°N, 26.5°N and 11°S are explored in respect to their ability to represent the AMOC and to monitor the temporal evolution

of the AMOC. Apart from the basin-scale measurements at 26.5°N, it is shown that in particular the outflow of North Atlantic Deepwater at 53°N is a good indicator of the subpolar AMOC trend during the recent decades, if the latter is provided in density coordinates. The good reproduction of observed AMOC and WBC trends in the most reasonable simulations indicate that the eddy-rich VIKING20X is capable in representing realistic forcing-related and ocean-intrinsic trends.

## 1   Introduction

The Atlantic Meridional Overturning Circulation (AMOC) is one of the most iconic quantities in large-scale oceanography and climate sciences (Srokosz et al., 2020; Frajka-Williams et al., 2019). As an integral calculation it summarises individual current systems and local velocities into a basin-scale latitude-depth representation. Owing to the combination of warm northward surface and cold southward deep flows, the AMOC is responsible for a net meridional heat transport from low to high latitudes (Biastoch et al., 2008a; Msadek et al., 2013), hence is a key for understanding the impact of the ocean on climate and the

evolution of climate change. Projections performed within the 'Climate Model Intercomparison Project' (CMIP, Eyring et al., 2016) foremost evaluate the future evolution of the AMOC strength (e.g. Weijer et al., 2020). And yet, despite its importance is the AMOC and its past evolution most difficult to obtain and to quantify.



Several attempts aim to monitor the AMOC. Building on historical measurements, RAPID at 26.5°N is the longest and most complete array, that continuously monitors boundary currents and interior geostrophy to combine with Ekman transports to a

full AMOC time series since 2004 (Moat et al., 2020). Others concentrate on individual currents where the AMOC manifests in individual surface or deep components, such as the western boundary current (WBC) structure in the Labrador Sea at 53°N (Handmann et al., 2018; Zantopp et al., 2017), the Line W off the U.S. coast (Toole et al., 2017), the MOVE array at 16°N (Send et al., 2011) or the North Brazil Current at 11°S (Hummels et al., 2015). As a basin-wide counterpart to RAPID, the SAMOC array aims to estimate the AMOC at 34.5°S (Garzoli and Matano, 2011; Meinen et al., 2018), but is available only

for shorter time periods and less complete because of the vigorous eastern and western boundary currents at this latitude. Owing to the importance of processes in the subpolar-subarctic North Atlantic, in particular for the decadal variations of the AMOC, most recent activities concentrate on a basin-wide array crossing the subpolar North Atlantic from both sides towards the southern tip of Greenland, covered through the international activity 'Overturning in the Subpolar North Atlantic Program' (OSNAP, Lozier et al., 2017), also including the array at 53°N . Common to all observational attempts is the limited spatial

and temporal coverage, that allows to focus only on individual components and/or limited time periods of up to a maximum of 24 years.

Numerical models help to expand the limited view from observations and guide the interpretation of the physical causes for the evolution of the AMOC. It has been shown that ocean general circulation models (OGCM) performed under past observed forcing, so-called 'hindcasts', simulate a robust and realistic interannual variability because of the direct impact of wind as a

driving force (Danabasoglu et al., 2016; Biastoch et al., 2008a). However, in particular the simulated decadal variability differs among individual model realisations because of the importance of deep water formation and spreading, processes that are very sensitive to choices of the numerics, resolution and parameterizations (Hewitt et al., 2020). Two specific aspects can be seen as instrumental for a proper simulation of the spatio-temporal evolution of the AMOC: an adequate ocean-grid resolution and a well-balanced atmospheric forcing.

Owing to the dominance of the mesoscale in the ocean, eddies play an important role, leading to the strong and fast changes of the AMOC seen on monthly and even daily time scales (Frajka-Williams et al., 2019). According to Hallberg (2013), a horizontal grid resolution of at least 1/10°, better 1/20°, is required to resolve the mesoscale in the subtropical and subpolar North Atlantic. An increased resolution of frontal and WBC structures also contributes to the correct simulation of pathways (Bower et al., 2019). As a specific aspect pertinent to simulations of the AMOC, it was also shown that the outflow of the

densest component of the North Atlantic Deep Water (NADW) through the Denmark Strait and the Faroe Bank Channel from the Nordic Seas, in particular, the entrainment of ambient water masses in the downslope flow regimes south of the sill, is strongly dependent on resolution and numerics (Legg et al., 2006).

For about 20 years basin-scale and global configurations exist at 1/10° or higher resolution. While early experiments aimed at realistically simulating WBC dynamics such as the separation of the Gulf Stream and eddy-mean flow interactions (Maltrud

et al., 1998; Smith et al., 2000; Eden and Böning, 2002), later studies concentrated on more challenging processes impacting the AMOC such as the convection and overflow (Treguier et al., 2005; Xu et al., 2010). The success of high-resolution models enabled detailed comparisons with the real ocean and improved the design and interpretation of ocean observations (Handmann





et al., 2018; Breckenfelder et al., 2017). Multi-decadal hindcasts can now be routinely integrated by a number of groups (Hirschi et al., 2020), also allowing to study the impact of external impacts such as enhanced melting from Greenland's glaciers (Böning et al., 2016).


Besides horizontal resolution, another important ingredient is a realistic atmospheric forcing. In contrast to coupled ocean-atmosphere models, which simulate intrinsic variability of the ocean circulation not necessarily in phase with observations, ocean hindcasts require a full set of atmospheric variables binding these to observed variability at the surface. The representation of the wind-driven circulation as well as of thermohaline-driven events depends on realistic representations of these

surface boundary conditions. An additional constraint to the availability of the specific data is a well-balanced set of variables for the heat and freshwater budgets. The atmospheric forcing data specifically created by Large and Yeager (2009) for the 'Coordinated Ocean Reference Experiments' (CORE, Griffies et al., 2009) was such a standard for the past decade. In recent years it was replaced by the new JRA55-do dataset, which is continuously updated to the present and available at higher spatial and temporal resolution (Tsujino et al., 2020). Forcing products for ocean models are limited by the lacking feedback between

the ocean and the planetary boundary layer in the atmosphere, e.g. through the inclusion of sea surface temperatures (SST) for the calculations of sensible heat flux and evaporation using Bulk formulae. However, in this formulation the atmospheric temperature needs to be prescribed and it cannot respond to changes in the SST, thereby attenuating an important negative feedback mechanism that in the real, coupled ocean-atmosphere system effectively acts to stabilise the AMOC (Rahmstorf and Willebrand, 1995). In consequence, the AMOC in these models can become more strongly influenced by the positive feedback

involved in the meridional freshwater transport, rendering them excessively sensitive to the freshwater forcing, e.g., to changes or errors in the prescribed precipitation and continental runoff (Griffies et al., 2009).

In this study, we describe an OGCM (VIKING20X) aiming at hindcast simulations of Atlantic Ocean circulation variability on monthly to multi-decadal time scales and with a spatial resolution sufficient to capture mesoscale processes well into subarctic latitudes. VIKING20X is an expanded and updated version of the original VIKING20 model configuration (Behrens,

2013; Böning et al., 2016), now covering the Atlantic from the Nordic Seas towards the southern tip of Africa with a 1/20°grid, nested into a global ocean/sea-ice model at 1/4° resolution. We demonstrate that both the 'eddy-rich' coverage and the new atmospheric forcing provide a configuration that improves the simulation of various key aspects of wind-driven and thermohaline ocean dynamics. However, we will also show that even at this resolution some numerical choices remain of critical importance, particularly for the evolution of the AMOC on inter-decadal and longer time scales. We will exploit this sensitivity here by

exploring a set of experiments differing in choices of the forcing (i.e., based on CORE and JRA55-do), initial conditions, and some aspects of the formulation of the freshwater fluxes. A particular emphasis of the study is on the imprints of AMOC variability and trends on WBC systems which, in turn, contributes to exploring the capability of regional observation systems to capture changes in the basin-scale AMOC. Using the different evolution of the experiments in respect to the long-term evolution of the AMOC, we turnaround the question and ask which regional observations are able to capture changes in the

AMOC.

This manuscript is organised as follows: After a comprehensive description of the model configurations and experiments and their atmospheric forcing (Section 2), we describe the basin-wide horizontal circulation and the AMOC (Section 3). Section 4




examines the regional representations of key components for the AMOC, from north to south. Section 5 discusses the result and summarises the manuscript.

## 2    Model configurations and atmospheric forcing

VIKING20X is an updated and expanded version of VIKING20. Originally developed by Behrens (2013) to study the impact of Greenland's melting glacier on the North Atlantic (Böning et al., 2016), hence representing the Atlantic Ocean from 32°N to 85°N at high resolution, VIKING20 has been shown to improve a series of key feature in the subtropical-subpolar North Atlantic compared to older and coarser-resolved models: the correct separation of the Gulf Stream and the subsequent path of the North Atlantic Current (Mertens et al., 2014; Breckenfelder et al., 2017; Schubert et al., 2018), the path of the Denmark Strait overflow into and around the subpolar gyre (Behrens et al., 2017; Handmann et al., 2018; Fischer et al., 2015), and the impact of the West Greenland Current eddies on the convection in the Labrador Sea (Böning et al., 2016). The success of the physical circulation enabled the use of VIKING20 also for a series of interdisciplinary applications such as the studies on the impact of ocean currents on the spreading of juvenile eels (Baltazar-Soares et al., 2014), the connectivity of deep-sea mussel populations (Breusing et al., 2016; Gary et al., 2020) and on the distribution of methanotrophic bacteria off Svalbard (Steinle et al., 2015). The ongoing use of VIKING20X for physical and biophysical studies motivates a complete model description and a thorough verification of the large-scale circulation.

VIKING20X has already been used by Rieck et al. (2019) to study mesoscale eddies in the Labrador Sea and their impact on the deepwater formation. They confirmed the ability of VIKING20X to simulate the generation of Irminger Rings (Brandt et al., 2004), convective eddies (Marshall and Schott, 1999) and boundary current eddies (Chanut et al., 2008) and their impacts, e.g., on the stratification of the Labrador Sea. In a model comparison on the AMOC Hirschi et al. (2020) found VIKING20X to be comparable with other eddy-rich models in respect to the representation of the AMOC. Rühs et al. (manuscript under review at *J. Geophys. Res.*) demonstrated a good representation of the amount and timing for deepwater formed in the Labrador Sea and showed that the model is capable in also simulating deepwater in the Irminger Sea.

### 2.1    ORCA025 and VIKING20X

The model simulations described and analysed in this study are based on the 'Nucleus for European Modelling of the Ocean' (NEMO, Madec, 2016) code version 3.6, also involving the 'Louvain la Neuve Ice Model' (LIM2, Fichefet and Morales Maqueda, 1997). The primitive equations describing the dynamic-thermodynamic state of the ocean are discretised on a staggered Arakawa C-type grid while the two-layer sea-ice model simulating one ice class with a viscous-plastic rheology is solved on a B-type grid. A global configuration (ORCA025) is used as an 'eddy-present' stand-alone configuration as well as host for the eddy-rich configuration VIKING20X where 'Adaptive Grid Refinement In Fortran' (AGRIF, Debreu et al., 2008) allows to regionally increase the resolution by embedding a so-called nest, here covering the Atlantic Ocean.

The global ORCA025 (Barnier et al., 2006) is described by orthogonal curvilinear, quasi-isotropic, tripolar coordinates yielding to a finer horizontal resolution with higher latitudes at a nominal grid size of 1/4°. The vertical grid is given by 46





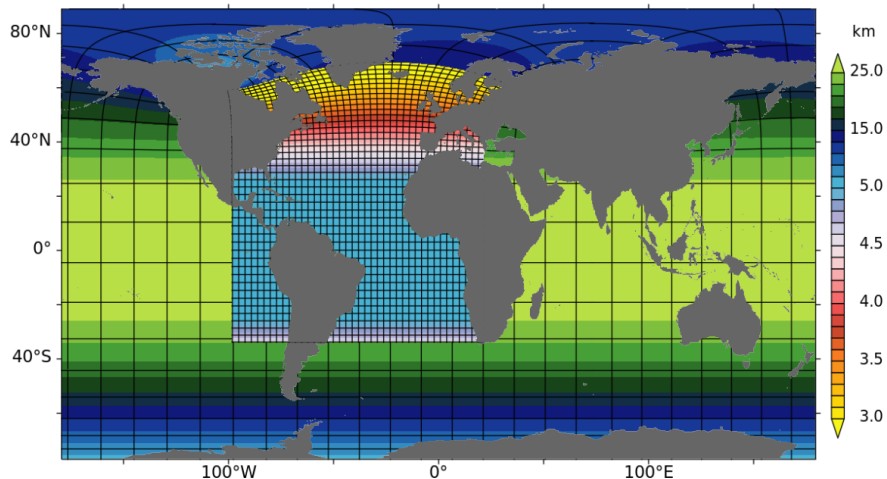

**Figure 1.** Domain and resolution (in km) of the VIKING20X configuration. The nest area is marked by increased resolution ranging from 5 to 3 km embedded into a global ORCA025 host grid (for both grids every 60th grid line is shown in $x$ and $y$ direction).

geopotential z-levels with layer thicknesses from 6 m at the surface gradually increasing to $\sim$250 m in the deepest layers. Bottom topography is represented by partially filled cells with a minimum layer thickness of 25 m allowing for an improved representation of the bathymetry (Barnier et al., 2006) and to adequately represent flow over the dynamically relevant $f/H$ contours ($f$ being the Coriolis Parameter and $H$ the water depth). Together with a momentum advection scheme in vector form with applied Hollingsworth correction (Hollingsworth et al., 1983), conserving both energy and enstrophy (EEN,

Arakawa and Hsu, 1990), this leads to an good representation of the large-scale, horizontal flow field (Barnier et al., 2006). For tracer advection, a 2-step flux corrected transport, total variance dissipation scheme (TVD, Zalesak, 1979) is used, ensuring positive-definite values. Momentum diffusion is given along geopotential surfaces in a bi-Laplacian form with a viscosity of $15 \times 10^{10}$ m$^4$s$^{-1}$. Tracer diffusion is along iso-neutral surfaces in Laplacian form with an eddy diffusivity of 300 m$^2$s$^{-1}$. Fast external gravity waves are damped applying a filtered free surface formulation (Roullet and Madec, 2000) in a linearised form

to ensure a volume conservative ocean. Horizontal sidewall boundary conditions are formulated as free-slip everywhere except for a region around Cape Desolation where no-slip is applied to improve the representation of West Greenland Current eddies (Rieck et al., 2019). A quadratic bottom friction term is applied as vertical boundary condition. In the upper ocean, a turbulent kinetic energy (TKE) mixed layer model (Blanke and Delecluse, 1993) diagnoses the depth of the mixed layer and increases vertical mixing for unstable water columns. This includes the representation of deep convection in formation regions of deep

and bottom waters.

VIKING20X consists of a global ORCA025 host grid and a nest covering the Atlantic Ocean from 33.5°S to $\sim$65°N (Owing to the tripolar grid, the northernmost latitude varies, with a maximum of 69.3°N and a mean of 65.1°N) at a nominal horizontal resolution of 1/20°. Both grids are connected through a two-way nesting capability, using AGRIF with a grid refinement factor of 5 (Fig. 1). Due to its mandatory rectangular shape on the host grid, the nest reaches into the Pacific Ocean to 100°W and





cuts through the Mediterranean at ∼22°E. (Note that the eastern boundary ranges from 20°E in the south to 32°E in the north.) Both grids share the same vertical axis. The bottom topography in the nest is generated by interpolating ETOPO1 (Amante and Eakins, 2009) to the model grid and connected to the host via a transition zone along the nest boundaries. Not only bathymetry but also coast lines are thereby better resolved. To guarantee that all ocean grid cells in the nest are embedded into 'wet' cells on the host grid, the coast lines on the host grid within the nested area are adjusted accordingly; an updated bathymetry is applied

for the host. To meet the Courant-Friedrichs-Lewy (CFL) criterion, the time step for the integration on the nest grid is reduced by a factor of 3 compared the host grid. The two-way nature of the nesting not only provides boundary conditions from the host to the nest but also communicates back the effect of resolving smaller scale processes in the nested area to the global ocean. This is achieved by an exchange along the nest boundaries in both directions at every common time step of the host and nest integration (here, every third nest time step). Furthermore, the solution on the host grid is updated with the three-dimensional

ocean state on the nest grid, usually every third host grid time step (for the freshwater budget corrected experiments, see below, this is done at every host grid time step).

Diffusion parameters are adjusted for the nest grid to meet the increased resolution. The Laplacian parameter for tracers is $60\,\mathrm{m^2 s^{-1}}$ and the bi-Laplacian parameter for momentum is $6\times10^9\,\mathrm{m^4 s^{-1}}$. To allow for a smooth transition between the host and the nest grids, a sponge layer is applied as a second-order Laplacian operator with a damping scale of $600\,\mathrm{m^2 s^{-1}}$.

## 2.2 Atmospheric forcing

For the simulations used here, we employ two different atmospheric forcing sets developed for the use in ocean models, CORE (version 2) (Large and Yeager, 2009; Griffies et al., 2009) and JRA55-do (Tsujino et al., 2020).

The CORE dataset is a merged product on a regular 2° grid covering the period 1948 to 2009. It builds on the NCEP/NCAR reanalysis which is corrected with observations and climatologies. Provided are zonal and meridional winds as well as air

temperature and humidity 10 m above sea level available as interannually varying fields at 6-hourly resolution throughout the entire forcing period. For the earlier phase, precipitation (at monthly resolution) and radiation (at daily resolution) are provided as climatology whereas for the later decades CORE incorporates precipitation (since 1979) and radiation (since 1984) as interannually varying fields from satellite-based measurements. Atmospheric fluxes are globally balanced on the basis of observed sea surface temperature and salinity data. A set of Bulk formulations also provided by Large and Yeager

(2009) connects the atmospheric forcing fields with the ocean model. The surface wind stress is formulated as relative winds, by using the difference between wind and ocean velocities for the calculation of the Bulk formulae. CORE was used for a series of ocean model intercomparisons (OMIP) (e.g., Griffies et al., 2009; Danabasoglu et al., 2014) and builds the basis for the official OMIP under CMIP6 (Griffies et al., 2016). For the simulation forced with the CORE dataset, we employ a monthly climatological field representing 99 of the major rivers and coastal runoff (Bourdallé-Badie and Treguier, 2006) to simulate

freshwater input from land to the ocean.

The CORE dataset is no longer maintained and the forcing period therefore ends with the year 2009. As a successor, JRA55-do (used here is version 1.4) is meant to replace CORE as a common forcing product for ocean hindcasts and for model inter-comparison studies (Tsujino et al., 2018). JRA55-do (with 'do' for 'driving the ocean') builds on the Japanese





reanalysis product JRA-55 with improvements through the implementation of satellite and several other reanalyses products.
All atmospheric forcing fields are available on a 1/2° horizontal grid at 3-hourly temporal resolution covering the period 1958 to 2019. JRA55-do will be continuously extended into the present at least until 2023. In the simulations presented here, the same Bulk formulations as used in CORE are applied.

Along with the atmospheric fields, JRA55-do also provides an interannually varying daily river runoff field at 1/4° horizontal resolution, which includes freshwater fluxes from ice sheets. For Greenland, this even includes the enhanced observed melting of the past decades (Bamber et al., 2018). This runoff field needs to be remapped from the JRA55-do to the ocean model grid. Here, the challenge is in the discrete placement of runoff along the different coastlines. The JRA55-do runoff covers a broader band along the coast while fjords and bays are differently represented on the two grids. The discontinuous nature of the runoff field prohibits a simple interpolation scheme. We thus created a remapping tool to reassign runoff to the model coastline preserving the spatial fine-scale heterogeneity of the forcing field. The runoff of each source grid node is conservatively redistributed within a radius of 55 km (80 km for VIKING20X-JRA-OMIP and ORCA025-JRA-OMIP) onto ocean nodes on the global (host) grid using a distance-weighted ($D^{-3}$) scheme to reduce spatial smoothing of the freshwater flux. A few forcing field nodes are located farther than 55 km from the model coastline, for instance far inside fjords not represented in the model's topography. We account for this error by proportional upscaling of the remapped global runoff field at each time instance. The rather exact remapping yields some high-runoff locations, such as the Amazon river mouth with a long term average discharge of 0.28 kg m$^{-2}$ s$^{-1}$. Runoff in VIKING20X-JRA-short and in VIKING20X-JRA-long before 1980 was applied including these locally very confined and high values, leading to some rare and only short lived instabilities. To overcome these, we apply a simple river plume scheme, i.e. a spreading of the runoff within a radius of 100 km, again applying distance-based ($D^{-1}$) weights to keep the focus on the actual river mouth for grid cells with at least 0.005 kg m$^{-2}$ s$^{-1}$ runoff, representing the 27 largest rivers in VIKING20X-JRA-long from 1980 onwards and in all other JRA55-do forced experiments. Compared to the previous runoff field used in conjuction with the CORE forcing these river plumes are considerably smaller. Runoff in the VIKING20X nest is then based on the runoff field on the host grid and interpolated onto the nest following the same procedure (Lemarié, 2006) as for all other initialisation fields. Inherent to this procedure is a spatial smoothing over 5×5 grid cells depending on the nesting scheme: runoff in the nest is supposed to enter the ocean in the same geographic area as it does on the underlying host grid, which has a five times coarser resolution. Note, the interpolation scheme erroneously assigns runoff to land grid cells, which we corrected by redistributing the runoff to ocean nodes within the associated 5×5 grid boxes.

### 2.3 Experiments

A series of simulations is used in this study (Table 1). VIKING20X-CORE is an experiment forced by the CORE dataset for the period 1958 to 2009, already used and described by Rieck et al. (2019) and Hirschi et al. (2020). It is based on a spin-up integration under the interannually-varing CORE forcing for the period 1980 to 2009 that originally started at rest from hydrographic conditions as provided by the World Ocean Atlas 1998 (Levitus et al., 1998) with corrections for the polar regions (PHC2.1, Steele et al., 2001). The sea-ice fields for the spin-up are initialised from a pre-spun state of a former simulation in ORCA025 to allow for a smooth start of the spin-up integration avoiding strong shocks to the fresh water budget.





**Table 1.** Experiments with forcing and integration period. Also provided are internal names used to identify the specific experiments. For the initialisation, 'Spinup' refers to an experiment covering the period 1980 to 2009 under CORE forcing, 'Rest' to an initialisation with temperatures and salinities of WOA13 and velocities at rest. VIKING20X-JRA-short was restarted from the end state of year 1979 of VIKING20X-CORE. SSSR is the sea surface salinity restoring timescale in m yr$^{-1}$, FWB is a potentially used freshwater budget correction. Experiments are grouped according to their initialisation and application of freshwater fluxes.

| Short name | Long/internal name | Forcing | Period | Initialisation | SSSR | FWB |
|---|---|---|---|---|---|---|
| VIKING20X-CORE | VIKING20X.L46-KKG36013H | CORE v2 | 1958-2009 | Spinup | 12.2 | - |
| VIKING20X-JRA-short | VIKING20X.L46-KKG36107B | JRA v1.4 | 1980-2019 | Year 1979 | 12.2 | - |
| VIKING20X-JRA-long | VIKING20X.L46-KFS001 | JRA v1.4 | 1958-2019 | Spinup | 12.2 | - |
| ORCA025-JRA | ORCA025.L46-KFS001-V | JRA v1.4 | 1958-2019 | Spinup | 12.2 | - |
| ORCA025-JRA-strong | ORCA025.L46-KFS006 | JRA v1.4 | 1958-2019 | Spinup | 50.0 | × |
| VIKING20X-JRA-OMIP | VIKING20X.L46-KFS003 | JRA v1.4 | 1958-2019 | Rest | 50.0 | × |
| ORCA025-JRA-OMIP (2 cycles) | ORCA025.L46-KFS003-V (-2nd) | JRA v1.4 | 1958-2019 | Rest | 50.0 | × |
| INALT20-JRA-long | INALT20.L464-KFS10X | JRA v1.4 | 1958-2019 | Spinup | 50.0 | - |

Three hindcasts in VIKING20X are forced by the JRA55-do forcing: VIKING20X-JRA-short is a short hindcast integration, branched off VIKING20X-CORE at the end of 1979 and performed from 1980 to 2019. VIKING20X-JRA-long is based on the
spinup for VIKING20X-CORE and performed over the whole forcing period 1958 to 2019. VIKING20X-JRA-OMIP instead follows the OMIP-2 protocol (Griffies et al., 2016) and started from rest and an initialisation of temperature and salinities of the World Ocean Atlas 2013 (WOA13, Locarnini et al., 2013; Zweng et al., 2013). It also differs in respect to the sea surface salinity (SSS) restoring and the balance of the freshwater budget (see below). For comparison, the two long-term experiments were accompanied by experiments in ORCA025: ORCA025-JRA and ORCA025-JRA-strong following VIKING20X-JRA-
long, and ORCA025-JRA-OMIP following VIKING20X-JRA-OMIP.

To reduce model drifts due to missing feedbacks from the atmosphere, a SSS restoring towards the initial climatological field is applied in most VIKING20X experiments with a piston velocity of 50 m 4.1 yr$^{-1}$ (12.2 m yr$^{-1}$) leading to a restoring timescale of 183 days for the uppermost 6-m grid cell. In sea-ice covered areas as well as where runoff enters the ocean, restoring is suppressed. Furthermore, at the river mouths vertical mixing in the upper 10 m of the water column is enhanced.
ORCA025-JRA-strong, ORCA025-JRA-OMIP and VIKING20X-JRA-OMIP instead used a stronger piston velocity of 50 m yr$^{-1}$ (timescale of 44 days) and a freshwater budget correction that globally balances the freshwater fluxes to zero at any host timestep. In all experiments under JRA55-do forcing restoring is also suppressed in an 80 km wide band around Greenland to allow for a free spread of the enhanced fresh water input to the ocean from melting ice-sheets.

For comparison, in particular to assess potential restrictions due to the location of the southern boundary in VIKING20X,
a nested configuration where the refinement from 1/4° to 1/20° applies to the South Atlantic and western Indian Ocean, is




used: INALT20-JRA-long, also performed under JRA55-do forcing similar to VIKING20X-JRA-long. A full description of INALT20 is provided by Schwarzkopf et al. (2019). In contrast to VIKING20X-JRA-long, the SSS restoring in INALT20-JRA-long is stronger ($50\,\mathrm{m\,yr^{-1}}$), and the restoring also applies around Greenland. INALT20-JRA-long is initialized with the ocean state of a spin-up integration in INALT20 under CORE forcing from 1980-2009. The lateral boundary condition

in INALT20-JRA-long is no-slip in the nest and free-slip on the host grid without any special treatment at Cape Desolation. Furthermore, INALT20-JRA-long includes the simulation of tides.

It is important to acknowledge that the integration history of eddy-rich models is often less systematic as one would like to have for a systematic evaluation, often aiming at the 'best' experiment under demanding computational costs. The use of accompanying experiments with ORCA025 helps to isolate individual choices, such as the SSS restoring parameter by

comparing ORCA025-JRA and ORCA025-JRA-strong. The latest experiment (VIKING20X-JRA-OMIP), also differing in the initialisation, is following the recent OMIP-2 protocol (Tsujino et al., 2020) and was completed during the writing of this manuscript. ORCA025-JRA-OMIP was already performed over a subsequently following second cycle through the JRA55-do forcing.

## 3    Basin-wide Circulation

We start the analyses with an evaluation of the basin-scale circulation. In contrast to the horizontal circulation, for which satellite altimetry provides a good estimate, there is no ground-truth for the general structure of the AMOC. Utilising the longest available observational time series by the RAPID Programme for an evaluation of the AMOC strength and evolution, we compare the different evolution of the experiments.

The broad patterns of the mean sea surface height (SSH) are similar in all experiments (Fig. A1), reflecting a robust repre-
sentation of the upper-layer circulation in the subtropical and subpolar gyres, the equatorial circulation, and the South Atlantic-Indian Ocean supergyre. For the path of the North Atlantic Current and the separation of the subtropical and subpolar gyres, VIKING20X shows a major improvement compared to ORCA025. The impact of resolution becomes even more apparent in the patterns of the SSH variability (Fig. 2). Gauged by the observational account provided by AVISO, the VIKING20X experiments show a much improved solution compared to ORCA025, applying both to the magnitude and to the horizontal patterns

of the mesoscale variability at the western boundary and along open-ocean currents such as the Azores Current at around 35°N. Prominent differences particularly concern the more realistic separation of the Gulf Stream near Cape Hatteras and the course of the North Atlantic Current in its northward turn into the Northwest Corner in VIKING20X compared to ORCA025.The latter represents an improvement also to its precursor version (VIKING20) that simulated a Northwest Corner extending too far north into the southern Labrador Sea (Breckenfelder et al., 2017).

More than the horizontal circulation, being in large parts already determined by the grid resolution and the wind field, the vertical overturning circulation strongly depends on both wind and thermohaline driving mechanisms. This does not only involve the applied atmospheric forcing itself but also details of its application such as SSS restoring and its impact on the freshwater budget at higher latitudes (Behrens et al., 2013). The AMOC, represented by the streamfunction derived from





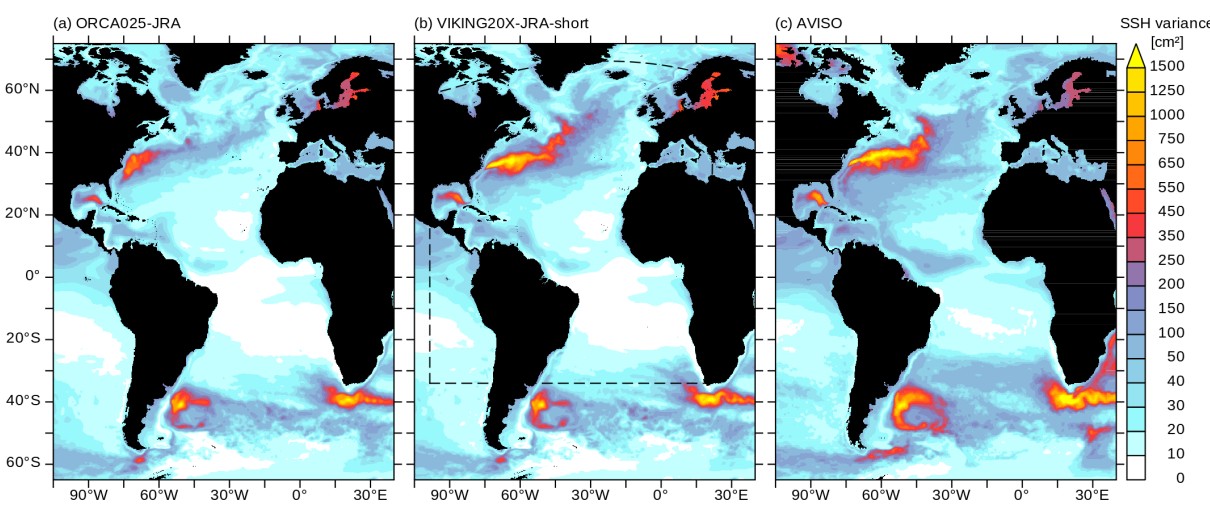

**Figure 2.** Variance of sea surface height (in cm$^2$) in (a) ORCA025-JRA, (b) VIKING20X-JRA-short and (c) satellite altimetry, calculated based on 5-day averages over the period 1993-2019.

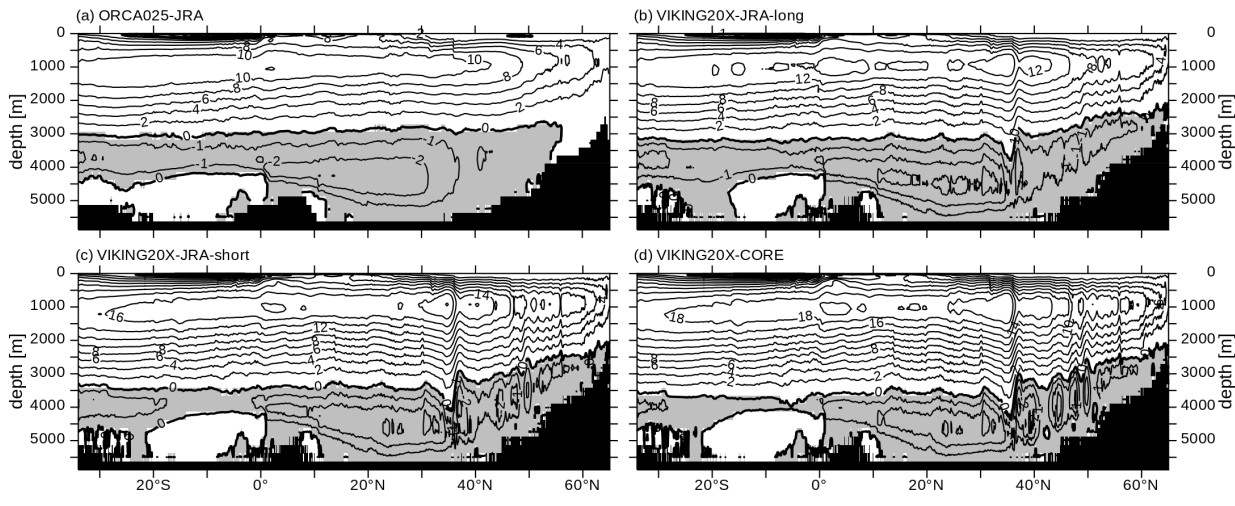

**Figure 3.** Mean AMOC streamfunction (1990-2009, in Sv) in (a) ORCA025-JRA, (b) VIKING20X-JRA-long, (c) VIKING20X-JRA-short and (d) VIKING20X-CORE. Positive (clock-wise) contour intervals are 2 Sv, negative (counter clock-wise; grey shaded) contour intervals are 1 Sv.

zonally and vertically integrated meridional velocities (Figure 3), reflects these influences in an integral way. While the general
structure, with the North Atlantic Deep Water (NADW) and Antarctic Bottom Water (AABW) cells, are broadly similar in





**Table 2.** Mean and standard deviation based on monthly averaged as well as interannually filtered (using a 23-months Hanning filter) data of the AMOC transport (given by the strength of the NADW cell) at 26.5°N. *Note the different period 2004-2009 due to the availability of RAPID data and shorter integration length of VIKING20X-CORE.

| Experiment | Mean [Sv] | | Std. dev. (2004-2018) [Sv] | |
| --- | --- | --- | --- | --- |
| | 1990-2009 | 2004-2018 | monthly | interannual |
| VIKING20X-CORE | 20.4 | 19.3* | 2.7* | 0.5* |
| VIKING20X-JRA-short | 18.0 | 15.3 | 3.0 | 1.1 |
| VIKING20X-JRA-long | 14.2 | 12.5 | 3.0 | 1.0 |
| ORCA025-JRA | 10.9 | 9.2 | 2.8 | 1.0 |
| ORCA025-JRA-strong | 13.4 | 12.6 | 2.7 | 0.7 |
| VIKING20X-JRA-OMIP | 18.3 | 15.9 | 2.9 | 1.1 |
| ORCA025-JRA-OMIP | 15.9 | 14.8 | 2.7 | 0.7 |
| ORCA025-JRA-OMIP-2nd | 14.2 | 13.4 | 2.7 | 0.8 |
| Observations (RAPID) | 18.6* | 17.7 | 3.4 | 1.5 |

all experiments, differences are apparent in both the strength and the vertical extensions of the NADW cells. The strength of the NADW cell is quite different, with 1990-2009 average values at 26.5°N ranging from 10.9 Sv in ORCA025-JRA to 20.4 in VIKING20X-CORE (Table 2). In the observational period (2004-2018), the JRA55-do-based experiments yield lower estimates, while VIKING20X-CORE (note the joint coverage of just 5 years) appears higher compared to observations at

26.5°N. Nevertheless, both VIKING20X-JRA-short and VIKING20X-JRA-OMIP fall well within the range of the observed interannual standard deviations. There is a clear resolution effect with ∼1-3 Sv higher transport in the 1/20° simulations depending on the exact comparison (VIKING20X-JRA-long vs. ORCA025-JRA or VIKING20X-JRA-OMIP vs. ORCA025-JRA-OMIP) and time period. The NADW cell is also deeper at high resolution, indicating a better representation of the lower component of the NADW constituted by overflow across the Greenland-Scotland Ridge and the corresponding entrainment of

ambient water (discussed further below). Finally, there is dependency of the mean AMOC on initial conditions, as illustrated by the different strength of ORCA025-JRA-strong and ORCA025-JRA-OMIP and its subsequent second cycle. Experiments with a longer history tend to simulate a weaker AMOC, pointing to a spin-down effect (see below). This in particular applies to VIKING20X-JRA-short that starts from a relatively high level of VIKING20X-CORE, thus simulates (over the same time period) a 3-4 Sv higher AMOC compared to VIKING20X-JRA-long under the same numerical conditions.

The RAPID data allow a more detailed evaluation of the depth structure. Figure 4a shows vertical profiles of the AMOC at 26.5° N; its vertical derivative represents the meridional transport per unit depth, thus providing a direct account of the northward and southward branches of the AMOC (Fig. 4b). Regarding the total strength of the NADW cell, all model results are lower than the observations (see also Table 2). Closer inspection shows that the differences mostly concern the representation of the deepest portion of the southward flow, i.e., the transport of lower NADW below ∼3200 m, whereas the upper part (1000-

3000 m) appears reasonably well represented. The deficit in the range of lower NADW has been recognised as a longstanding,

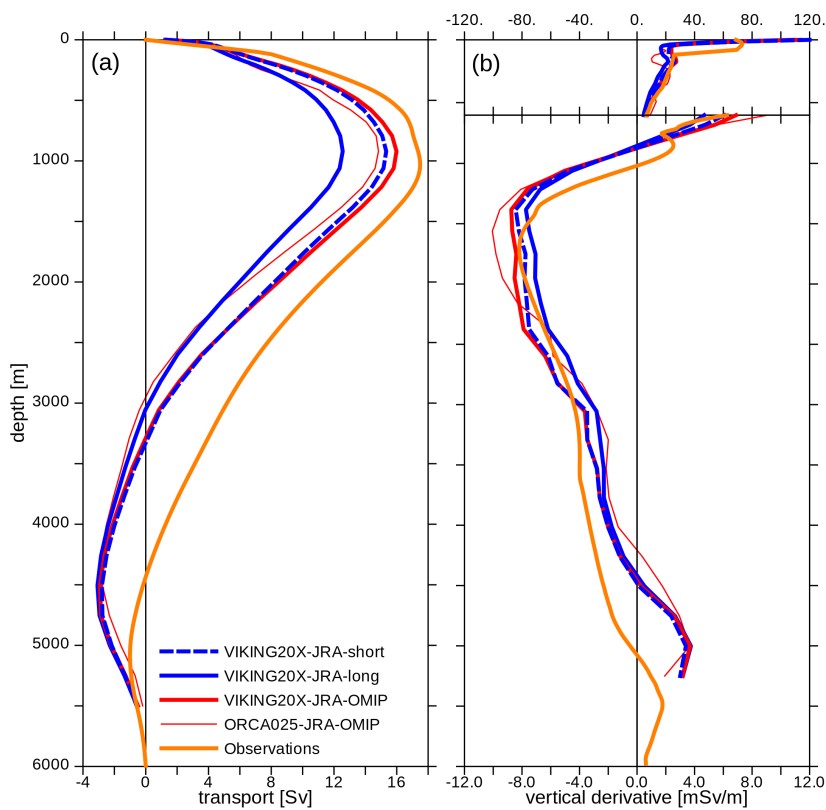

**Figure 4.** Profiles of the (a) AMOC (in Sv) and (b) its vertical derivative (in mSv m$^{-1}$) at 26.5°N (note the different ranges above and below 600 m depth) for experiments and RAPID observations (orange), all averaged from 2004 to 2018.

persistent issue in ocean and climate models (Fox-Kemper et al., 2019), and can largely be attributed to a loss of the high-density source waters from the Nordic Seas, e.g., by spurious mixing in the outflows across the Greenland-Scotland ridge system (Legg et al., 2006). The deficit is most pronounced in ORCA025; the representation is improved in VIKING20X, but there is still a gap by about 500 m in the reversal from southward NADW to northward AABW flow (Figure 4b). It remains unclear if this is a result of a too weak representation of the densest NADW, e.g. through spurious entrainment into the overflow, or by a too strong modelled AABW cell. It could also be influenced by the choice of the reference level used for the RAPID array (Sinha et al., 2018), noting that the representation of AABW is not its major aim.

Figure 5a shows that the AMOC differs not only in mean strength but also exhibits pronounced differences in its temporal evolution over multi-decadal time scales. The CORE-based experiment (VIKING20X-CORE) produces an increasing AMOC with a maximum in the mid-1990s, and a decline and stabilisation thereafter. This maximum corresponds well to the reported phase of strong convection in the Labrador Sea from observations (e.g. Yashayaev, 2007). While VIKING20X-JRA-long

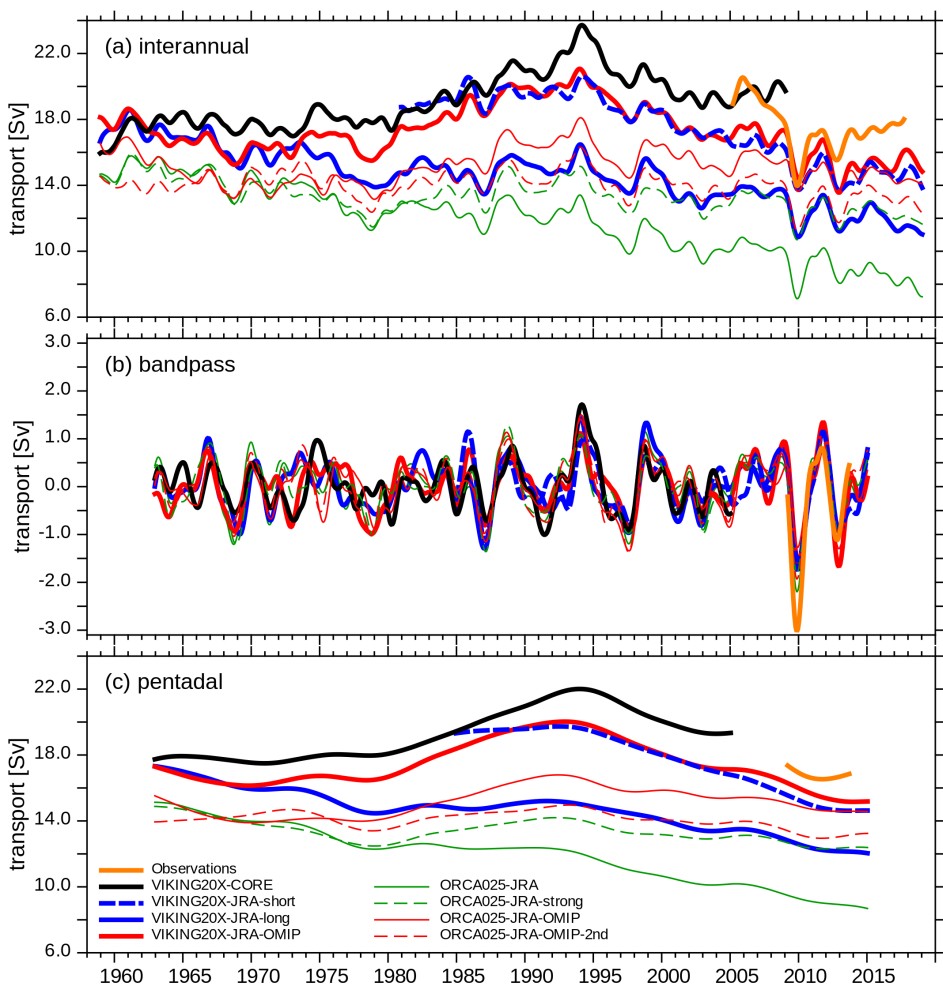

**Figure 5.** AMOC evolution, provided by the strength of the NADW cell at 26.5°N: (a) full interannual, (b) 1-5-year band-pass and (c) pentadally filtered time series.

exhibits a long-term decline, VIKING20X-JRA-OMIP also simulates the maximum in the mid-1990s, however in contrast to VIKING20X-CORE continues to decline in the 2000s and beyond. The observational period of RAPID is only fully covered by the JRA55-do-based experiments. If we consider VIKING20X-JRA-OMIP and VIKING20X-JRA-short showing the 'best'

evolution, we note a weaker AMOC at the beginning of the observational time series, but a good representation of the 2010 minimum which was described as a wind-related response in a negative NAO winter (e.g., McCarthy et al., 2012). There is a tendency towards a recovery thereafter, though with a stabilisation at a weaker level, compared to the observations, towards the end of the time series.


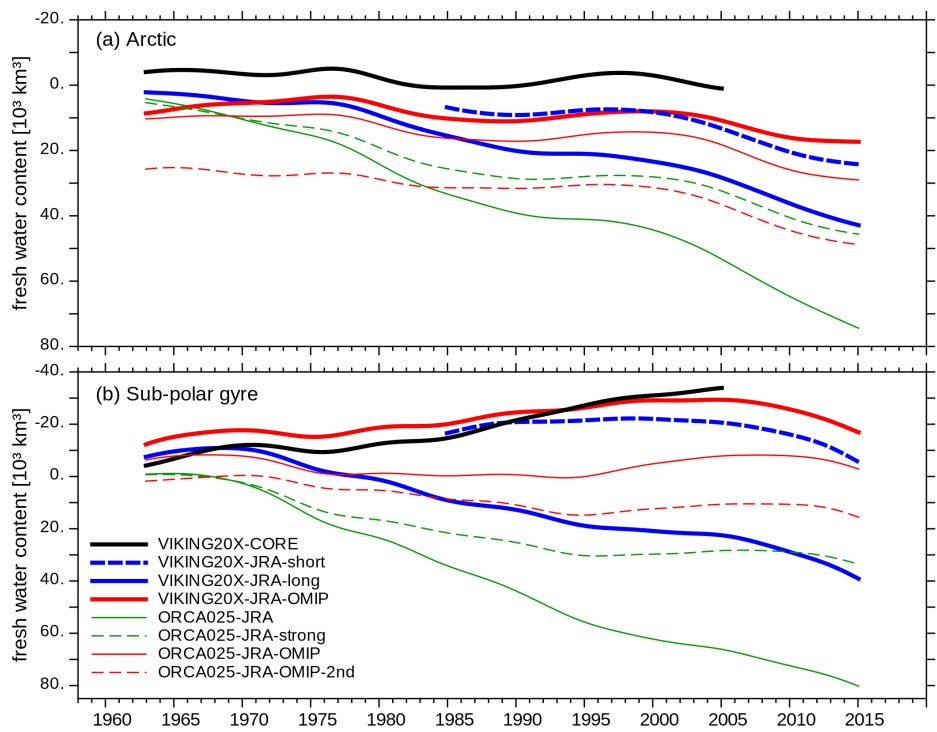

**Figure 6.** Pentadally filtered (a) Arctic and (b) subpolar Freshwater Content (computed from seawater alone, hence excluding sea-ice and snow volume, using a reference salinity of $S_{ref} =$34.7, in $10^3$ km$^3$). Note the inverted y-axis.

The interannual AMOC variability is remarkably robust among the range of experiments and in comparison to the observa-
tions (Figure 5b). The interannual correlations between VIKING20X-JRA-long, ORCA025-JRA and ORCA025-JRA-strong
range around $r = 0.8 - 0.9$ (10980-2009). VIKING20X-JRA-long correlates weaker with VIKING20X-CORE ($r = 0.6 - 0.7$),
probably because of the different wind forcing. Within the overlapping period, all experiments show a good correlation with
RAPID observations ($r = 0.73 - 0.86$). This underlines the importance of the wind forcing as a major driver for the interannual
variability (Danabasoglu et al., 2016). It is interesting to note that the observed monthly variability (indicated by the standard
deviation in Table 2) is underestimated by 10-20% with only little resolution dependency. However, the interannual variability
of VIKING20X is higher than that of ORCA025, but still underestimates the observations by more than 30%. We also no-
tice that the high variability of the latter may include errors from measurements and the processing of the different AMOC
components from RAPID data.

An important aspect for the long-term evolution of the AMOC is freshwater fluxes provided by the atmospheric forcings
and the numerical details of its application. This is demonstrated by also considering the ORCA025 sensitivity experiments:



ORCA025-JRA-strong (dashed green lines in Fig. 5a,c) with stronger SSS restoring and applied freshwater budget correction shows a weaker trend (at least post-1980s) compared to ORCA025-JRA (solid green lines). This is also indicated by VIKING20X-JRA-long and VIKING20X-JRA-OMIP (thick solid blue and red lines in Fig. 5a,c), although an initialisation effect with one experiment starting from a restart, the other from rest, could also play an additional role here.

The long-term evolution, stable and upward in VIKING20X-CORE and downward in the JRA55-do-based experiments (Fig. 5c), can be understood through the inspection of the evolution of the freshwater content (FWC). The trend of the Arctic FWC over the last couple of decades is stable in VIKING20X-CORE and VIKING20X-JRA-OMIP, and increasing in the other JRA55-do forced experiments (Fig. 6a — note the reversed y-axis to match to Fig. 5c). The trend has its origin in an increased precipitation within the Arctic and sub-Arctic regions, which also causes a slight increase in the river runoff. It fits

to the observed increase of $600 \pm 300$ km$^3$ year$^{-1}$ between 1992 and 2012 found by Rabe et al. (2014).

  In contrast to the Arctic, the subpolar North Atlantic shows a different evolution in the JRA55-do based experiments, depending on the application of the freshwater fluxes, and in consequence less of the forcing data itself (Fig. 6b). This effect is isolated by the two ORCA025 experiments: While ORCA025-JRA shows strong increases in FWC, the experiments with a stronger SSS restoring and freshwater budget correction, ORCA025-JRA-strong and ORCA025-JRA-OMIP, stabilise af-

ter 1980. As a result of the inability of Bulk formulae to properly feedback to the (largely prescribed) atmosphere, Griffies et al. (2009) have described a positive feedback between AMOC strength and freshwater forcing, with additional freshwater in the subpolar North Atlantic limiting deepwater formation. The corresponding reduction of the AMOC would cause less salt transported northward, in consequence leading to a further freshening of the subpolar North Atlantic (Behrens et al., 2013). Both VIKING20X-JRA-OMIP and ORCA025-JRA-OMIP seem to minimise the positive feedback between subpolar North

Atlantic FWC and AMOC through stronger SSS restoring and a global FWC correction obviously lowering the FWC trend in the subpolar gyre.

  While a clear attribution of the AMOC trends to either physical drivers (i.e., atmospheric forcing and runoff) or spurious model drift is not possible at this stage, we can use the range of solutions with their diverging trends to assess their manifestation in regional current systems, and thereby explore if and how regional observational arrays may be capable in depicting the long-

term evolution and variability of the AMOC. An important part of the analysis is the formation and spreading of deepwater masses. From a number of observational and modelling studies, Lozier (2010) concluded that NADW only partly follows a coherent Deep Western Boundary Current (DWBC) as explained by classical theory (Stommel, 1958). In several parts of the Atlantic Ocean deviations into the interior, recirculations and disruptions by deep mesoscale eddies play an important role in the spreading.

Figure 7 illustrates the pathways of NADW, entering from the Nordic Sea through the Denmark Strait and through the Faroe Bank Channel, the latter flowing around and crossing through gaps of the Reykjanes Ridge (Zou et al., 2017). The coherent path around Greenland is broken into mesoscale eddies in the northern Labrador Sea reaching even down into this density range (Rieck et al., 2019), and re-configures again at the Canadian side. The export of NADW from the subpolar into the subtropical North Atlantic and further into the South Atlantic is subject to many studies (e.g. Bower et al., 2009). General consensus,

indicated by models and observations is that a large part of NADW is deviated on a broad path towards the mid-Atlantic ridge



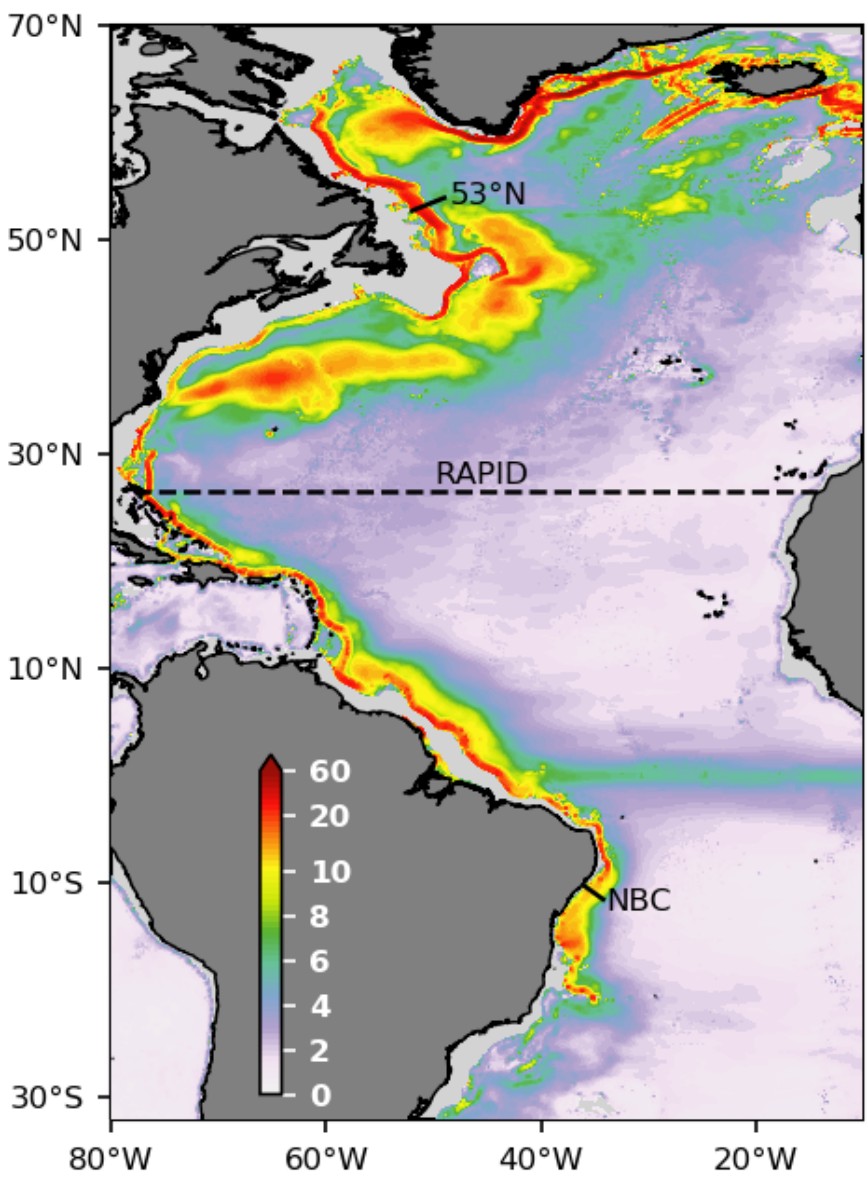

**Figure 7.** Mean speed (1990-2009, in cm s$^{-1}$), averaged between $\sigma_0 = 27.65 - 27.95$ in VIKING20X-JRA-short. Regional sections are indicated by black lines.

(Lozier et al., 2013; Gary et al., 2011; Biló and Johns, 2019; Le Bras et al., 2017), with only a narrow portion of the DWBC flowing around Flemish Cap and through Flemish Pass. Only at around 30°N, the flow towards the south is seen again as a coherent DWBC, but also subject to local recirculations (Schulzki et al., manuscript under review at *J. Geophys. Res.*). In the




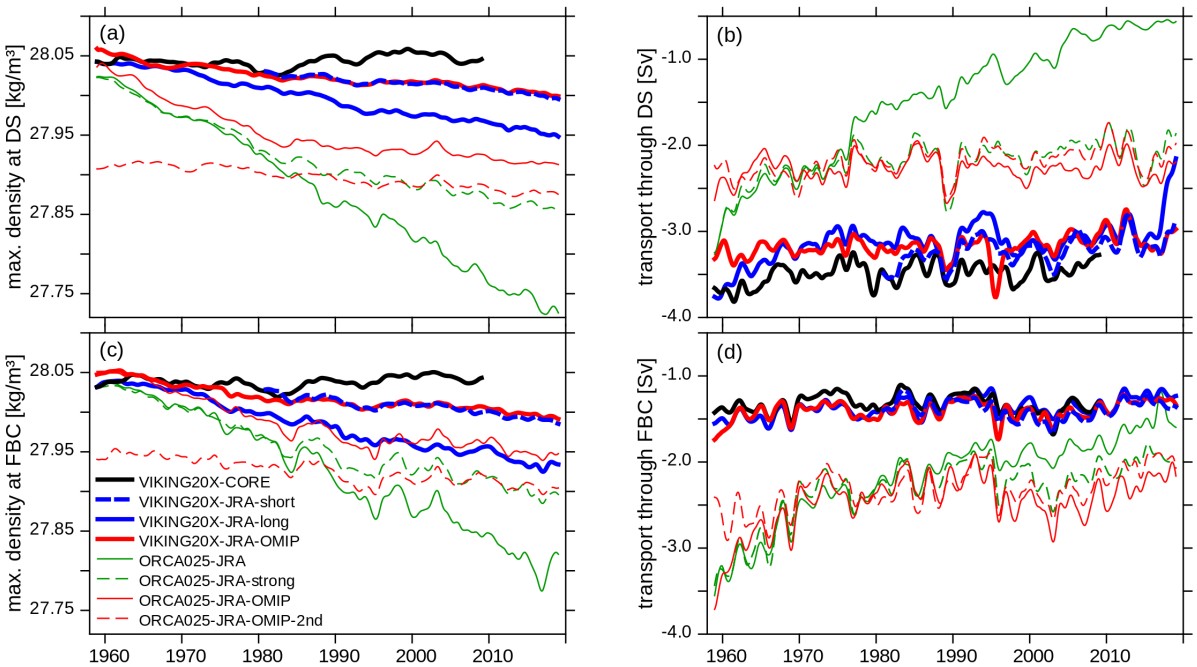

**Figure 8.** Evolution of maximum overflow density and the overflow transport in the (a and b) Denmark Strait and the (c and d) Faroe-Bank Channel respectively. Overflow transport estimates are based on southward transport of waters with density larger than 27.70 kg m$^{-3}$ below 270 m.

South Atlantic, the DWBC again breaks up into mesoscale eddies (Dengler et al., 2004), and then fades out at around 20°S.

(Van Sebille et al., 2012) has described the flow of NADW as a zonal confined pathway at 25°S. This is not seen here.

# 4  Regional Imprints

## 4.1  Subpolar North Atlantic

The subpolar North Atlantic is a key region for the AMOC. It receives surface water masses from the subtropics and overflow water from the Nordic Seas. Here, the different components of the NADW are formed through exchange with the atmosphere

and mixing processes. They directly maintain the strength of the AMOC and modulate its variability.

The densest component of the NADW is formed in the Nordic Seas: through heat loss to the atmosphere and sea-ice formation, dense water is formed and, by convection, builds a large reservoir at depth between Greenland, Iceland and Norway. It then spills over the Greenland-Scotland Ridge into the subpolar North Atlantic. Two narrow passages, the Denmark Strait between Greenland and Iceland with a sill depth of 650 m and the Faroe Bank Channel between the Faroe Islands and Scotland





**Table 3.** Maximum density and overflow transports provided as mean and monthly standard deviation for the period 1990-2009 at Denmark Strait sill and Faroe Bank Channel. Overflow transports are based on southward transport of waters with density larger than 27.70 kg m$^{-3}$ below 270 m.

| | max density $\sigma_0$ [kg m$^{-3}$] | | overflow transport [Sv] | |
| --- | --- | --- | --- | --- |
| | Denmark Strait | Faroe Bank Channel | Denmark Strait | Faroe Bank Channel |
| VIKING20X-CORE | 28.05 | 28.04 | $3.4 \pm 0.5$ | $1.4 \pm 0.2$ |
| VIKING20X-JRA-short | 28.01 | 28.01 | $3.2 \pm 0.5$ | $1.4 \pm 0.2$ |
| VIKING20X-JRA-long | 27.98 | 27.96 | $3.1 \pm 0.4$ | $1.3 \pm 0.2$ |
| ORCA025-JRA | 27.82 | 27.88 | $1.0 \pm 0.3$ | $1.9 \pm 0.2$ |
| ORCA025-JRA-strong | 27.89 | 27.93 | $2.1 \pm 0.5$ | $2.2 \pm 0.4$ |
| VIKING20X-JRA-OMIP | 28.02 | 28.01 | $3.2 \pm 0.5$ | $1.4 \pm 0.3$ |
| ORCA025-JRA-OMIP | 27.93 | 27.96 | $2.3 \pm 0.5$ | $2.4 \pm 0.6$ |
| ORCA025-JRA-OMIP-2nd | 27.89 | 27.91 | $2.1 \pm 0.5$ | $2.3 \pm 0.5$ |

with a sill depth of 850 m, funnel this exchange. Figure 8 and Table 3 show density and transport through both passages. With transports of around 3 Sv and little variability through the Denmark Strait, the mean transport in VIKING20X fits to the observational estimates (3.1 Sv by Jochumsen et al. (2017) and 3.5 Sv by Harden et al. (2016)). Transports through the Faroe Bank Channel are around 1.4 Sv, thus smaller than the reported 2.2 Sv (Hansen et al., 2016; Østerhus et al., 2019; Rossby et al., 2018) to 2.7 Sv (Berx et al., 2013). ORCA025 instead simulates an enhanced transport which (in parts) can be attributed to the

40% larger cross section at 1/4° resolution compared to 1/20°. The simulated maximum density is typically smaller than the reported $\sigma_0$=28.05-28.07 (Harden et al., 2016; Hansen et al., 2016), which can also be due to the limited vertical resolution not resolving the bottom boundary layer.

Except for a strong weakening trend in ORCA025-JRA and a spindown in the first decades of ORCA025-JRA-OMIP, transports do not show a long-term trend, and are quite stable. This is probably a result of the continuous supply of dense

water north of the sills and hydraulic control limiting the transport to its given value (Käse et al., 2003). More important than the transport itself is the density of the overflow water. Previous studies described a direct link between overflow density and AMOC strength (Behrens et al., 2013; Latif et al., 2006), although the exact reason for this is still unclear and debated. For example, Danabasoglu et al. (2014) do not find such a link in the variety of CORE-based experiments. Here we do see a similar behaviour of both overflow density and AMOC, with stable densities in VIKING20X-CORE and (different) weakening

trends in VIKING20X-JRA-long, VIKING20X-JRA-short and VIKING20X-JRA-OMIP. ORCA025-JRA shows a decline in overflow densities, stronger than anticipated from the AMOC trend (Figure 5). However, similar to the AMOC, ORCA025-JRA-strong shows a weaker declining trend compared to ORCA025-JRA after 1980. In ORCA025-JRA this is also reflected in the overflow transport. It is interesting to note that the first cycle in ORCA025-JRA-OMIP shows a similar stabilisation after about 25 years. The second cycle continues the remaining weak trend, but starts from a lighter density which is again reflected

in the AMOC. Instead, VIKING20X-JRA-OMIP does not show such a spin-down.


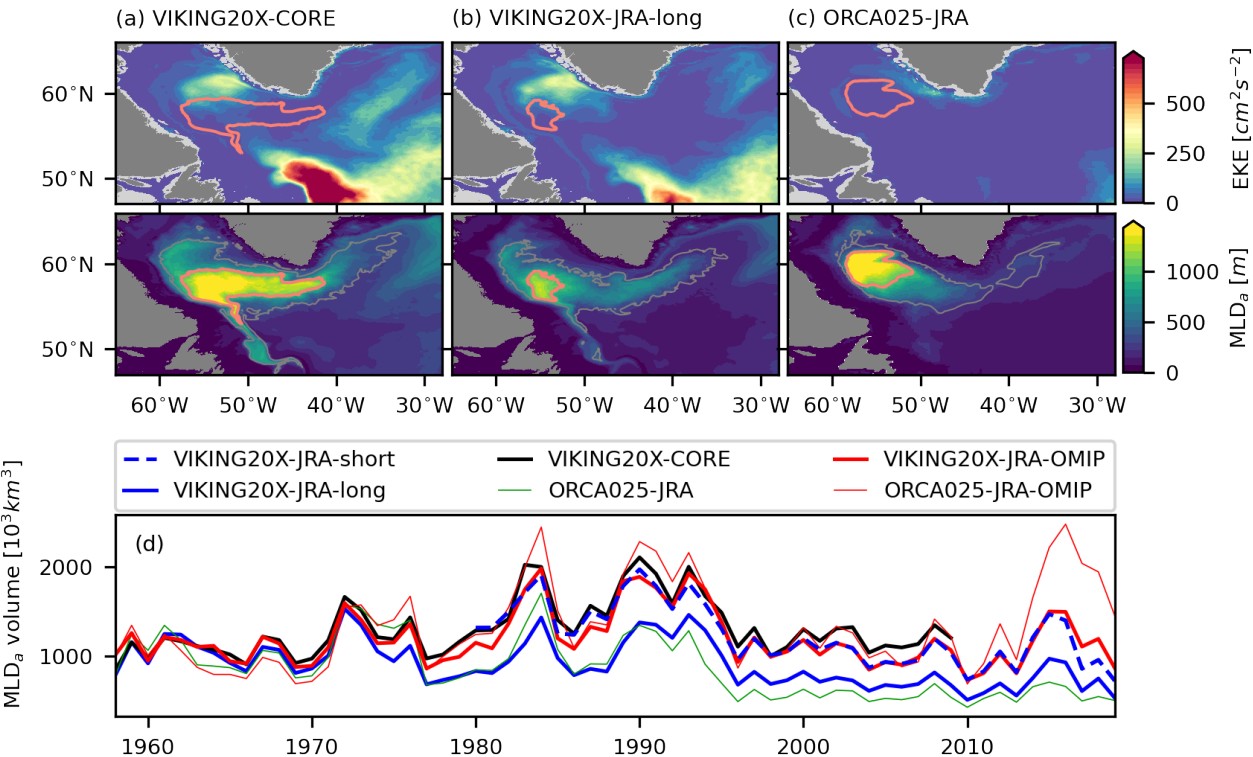

**Figure 9.** Spatial pattern and temporal variability of MLD in the subpolar North Atlantic: Long-term (1980-2009) mean EKE at 112 m depth and annual maximum MLD ($MLD_a$) in (a) VIKING20X-CORE, (b) VIKING20X-JRA-long and (c) ORCA025-JRA, light red contours highlight long-term mean $MLD_a > 1000$ m and grey contours the long-term maximum $MLD_a > 1000$ m; (d) interannual variability of spatially integrated $MLD_a$ volume.

In the subpolar North Atlantic, additional deepwater is added to the system. Owing to strong wintertime heat loss, facilitated among others through strong and cold winds, the Labrador and Irminger Seas are regions of deepwater formation through deep convection, providing a lighter, upper component to the deepwater, the upper NADW (in contrast to the overflows named lower NADW).

The distribution of long-term mean annual maximum mixed layer depth ($MLD_a$, Fig. 9a-c) shows that the spatial patterns of deep convection are influenced by both the ocean model resolution and the atmospheric forcing. In VIKING20X simulations the centre of deep convection in the Labrador Sea, here indicated by the light red line encompassing the area where long-term mean $MLD_a$ exceeds 1000 m, is limited in the north through the impact of travelling Irminger Rings visible through a tongue of elevated EKE (Fig. 9a-b, note that the pattern for VIKING20X-JRA-short and VIKING20X-JRA-OMIP are not shown but





are very similar to VIKING20X-CORE) as thoroughly described by Rieck et al. (2019). As Irminger Rings are not properly represented in ORCA025, the centre of deep convection here extends further to the northwest (Fig. 9c). Moreover, in all model simulations, the potential deep convection region, here indicated by the light grey line encompassing the area where long-term maximum $MLD_a$ exceeds 1000 m, extends into the Irminger Sea. However, the area covered by the centre of deep convection as well as by the potential deep convection region vary among the different model simulations, with no clear relation to the

model resolution. While there are little differences between VIKING20X-CORE, VIKING20X-JRA-short and VIKING20X-JRA-OMIP (not shown), VIKING20X-JRA-long and ORCA025-JRA feature overall smaller areas and ORCA025-JRA-OMIP (not shown) larger areas than the former.

While the resolution seems to determine the general spatial structure, the forcing and other model specific settings impact the intensity and temporal variability of deep convection. During the first 15 years, i.e., until the mid 1970s, the $MLD_a$ volume in

the depicted domain (Fig. 9d) shows nearly the same magnitude and temporal variability for all simulations (notably, the overall MLD volume in the ORCA025 simulations is not systematically larger than in the VIKING20X simulations). Afterwards, $MLD_a$ volume and variability in VIKING20X-JRA-long and ORCA025-JRA decreases compared to the other simulations. The smaller $MLD_a$ volume is a result of shallower $MLD_a$ over the whole domain, including reduced convection intensity in the central deep convection areas. Most interestingly, in VIKING20X-JRA-long and ORCA025-JRA the decrease of $MLD_a$

volume sets in after the simulated AMOC decline described above, suggesting that the long-term AMOC decline is not triggered by weakening deep convection in the subpolar North Atlantic, but potentially adds to the decrease in deep convection intensity (which, however, could then feedback on the AMOC). Hence, at least part of the diagnosed negative $MLD_a$ volume trends in VIKING20X-JRA-long and ORCA025-JRA arise from spurious model drifts described above.

In comparison to observations the $MLD_a$ patterns in the VIKING20X simulations (including the occasionally large MLDs

in the Irminger Sea) seem more realistic than that of other model simulations at lower or comparable resolution. Moreover, the temporal evolution of the MLD volume in VIKING20X-CORE, VIKING20X-JRA-short, and VIKING20X-JRA-OMIP agrees very well with the reported history of deep convection in the subpolar North Atlantic (while VIKING20X-JRA-long and ORCA025-JRA seem to miss a clear maximum of MLD volume and deep convection intensity in the 1990s, and ORCA025-JRA-OMIP experiences a too strong intensification of deep convection in recent years). A more detailed evaluation and inter-

pretation is done elsewhere (Rühs et al., manuscript under review at *J. Geophys. Res.*).

The circulation in the subpolar North Atlantic can be characterised by an index based on sea surface height. Following Koul et al. (2020), the subpolar gyre index is defined as 2nd principle component of EOF analysis with non-detrended data. It is highly correlated with steric changes in the gyre, e.g., upper ocean density in the centre of the gyre, which are largely connected to changes in the NAO index, and impact the upper ocean salinity in the eastern subpolar North Atlantic. A density increase in

the centre intensifies the gyre through geostrophic balance (both, stronger and larger gyre, index $> 0$), reduces the throughput of subtropical waters, and hence leads to a freshening of the eastern subpolar North Atlantic.

Figure 10 shows that the annual mean subpolar gyre index of most experiments is correlated with observations (Table 4). The experiments seem also robust on longer, decadal timescales, with a strong subpolar gyre in the 1990s, a weakening thereafter and a recovery in the mid 2010s. While both maxima are reflected in the convection strength (the second one at least





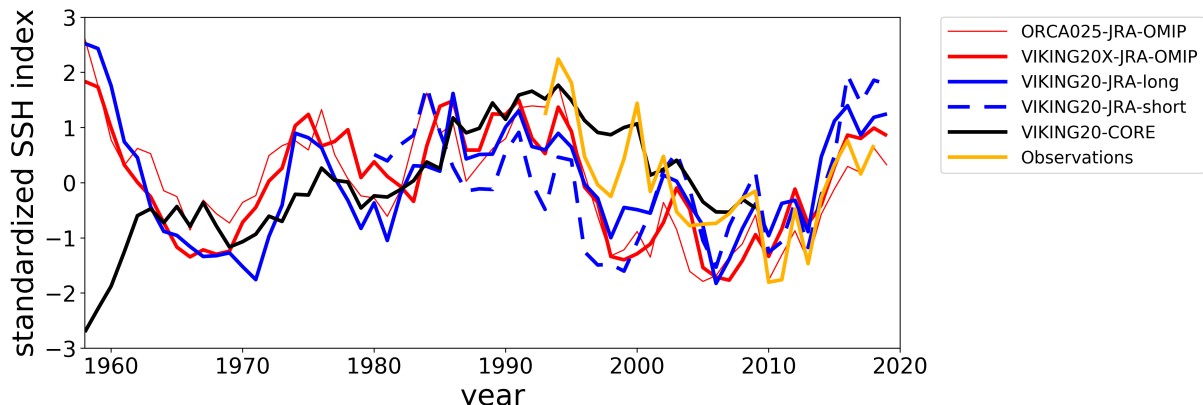

**Figure 10.** Time series of the annual mean subpolar gyre index in the experiments and based on observations (orange). Here, the index is defined as the second PC of an EOF-analysis for non-detrended SSH in the North Atlantic between 20 and 70°N.

**Table 4.** Correlations of the annual mean gyre index with observations for the period 1993-2009. Correlations are significant at 99%, except the one for VIKING20X-JRA-short (80%).

| Experiment | Correlation |
|---|---|
| VIKING20X-CORE | 0.85 |
| VIKING20X-JRA-short | 0.34 |
| VIKING20X-JRA-long | 0.76 |
| VIKING20X-JRA-OMIP | 0.71 |
| ORCA025-JRA-OMIP | 0.86 |

in VIKING20X-JRA-short, VIKING20X-JRA-OMIP, and ORCA025-JRA-OMIP see Fig. 9), the AMOC only represents the one in the 1990s (Fig. 5).

One important key location picking up the different constituents and timescales of subpolar gyre variability and deepwater formation is the observational array off Labrador at 53°N (Zantopp et al., 2017). The DWBC at this location is seen as an index for the subpolar AMOC and for the overall AMOC on decadal and longer timescales due to the increasing meridional coherence

(Böning et al., 2006; Bingham et al., 2007; Wunsch and Heimbach, 2013; Buckley and Marshall, 2016, see also discussion). Since 1997 this mooring array has recorded all three constituents of the NADW exiting the Labrador Sea via the DWBC (Zantopp et al., 2017; Fischer et al., 2004). The observations revealed a 100-150 km wide well-defined cyclonic (southward) boundary current, featuring a strong barotropic component with significant baroclinic flow in the shallow velocity maximum of the Labrador Current and the deep velocity maximum (typically 0.25 m s$^{-1}$, reaching up to 0.4 m s$^{-1}$) associated with the

lower NADW, and an anticyclonic recirculation in the interior Labrador Sea (Fischer et al., 2004; Lavender et al., 2000). Figure 11 a-c show that the general structure of the narrow boundary current system is visible especially in VIKING20X. Similarly

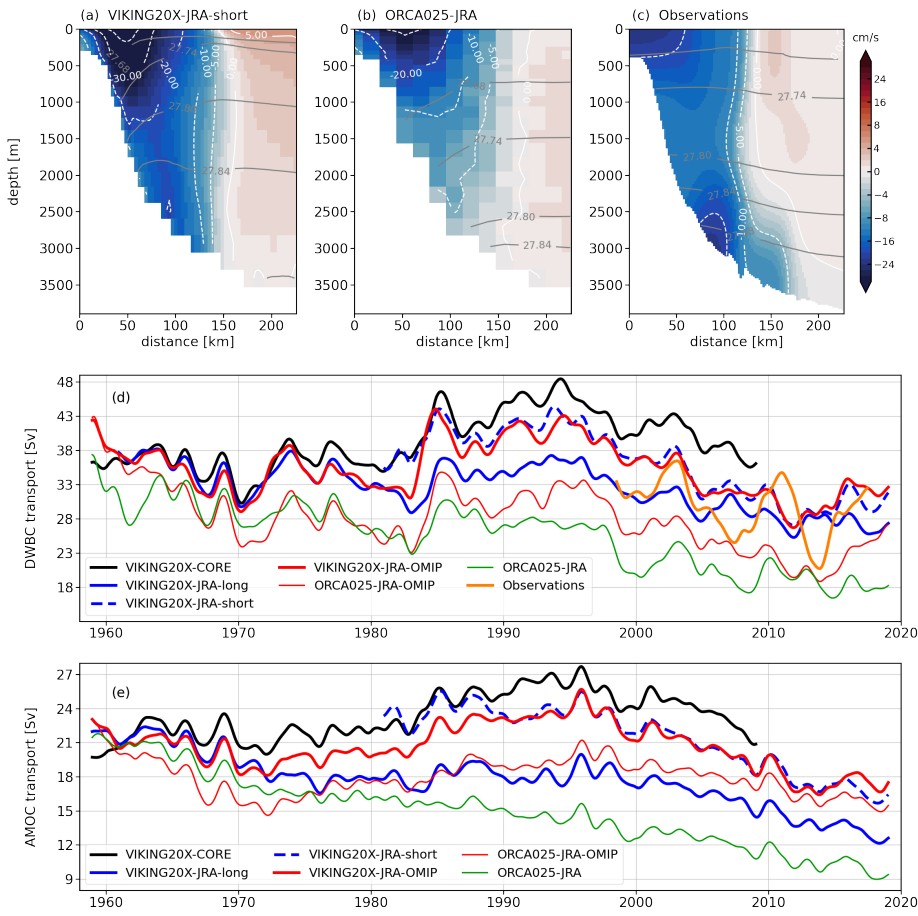

**Figure 11.** Mean sections (upper panel) of velocity and $\sigma_0$ isolines (1997-2009) at 53°N in (a) VIKING20X-JRA-short, (b) ORCA025-JRA and (c) observations (non-linear colour map is used with intervals given of 1 Sv between -8 and 8 Sv, and 2 Sv beyond that range. Time series of (d) the DWBC export across 53°N characterised by the NADW defined by $\sigma_2$ criteria (Table 5), following the analysis of Handmann et al. (2018) and (e) the AMOC transport in $\sigma_2$ density coordinates at 53°N .

to experiments with the predecessor VIKING20, they produce a stronger surface maximum, a weaker deep velocity maximum and a stronger recirculation than in observations (Handmann et al., 2018). In the ORCA025 experiments the boundary current appears too wide with a split surface maximum and no deep boundary current core, the latter pointing to a too strong erosion of lower NADW on its way around the Irminger and Labrador Seas. Both, the ORCA025-JRA and the VIKING20X-JRA-short are






**Table 5.** DWBC export (mean and std.dev.) across 53°N for different periods, characterised by the NADW defined by $\sigma_2$ criteria (ORCA025-JRA $> 36.52$, ORCA025-JRA-strong $> 36.63$, VIKING20X-JRA-long $> 36.59$, VIKING20X-JRA-short $> 36.68$, VIKING20X-CORE $> 36.72$, ORCA025-JRA-OMIP $> 36.73$ and VIKING20X-JRA-OMIP $> 36.69$), and 20-year trends.

| Experiment | Transport (1998-2009) | Transport (2000-2017) | Trend (1998-2017) |
|---|---|---|---|
| VIKING20X-CORE | $39.7 \pm 5.7$ Sv | $39.6 \pm 5.1$ Sv* | -0.48 Sv/yr* |
| VIKING20X-JRA-short | $34.6 \pm 4.7$ Sv | $32.0 \pm 4.9$ Sv | -0.50 Sv/yr |
| VIKING20X-JRA-long | $30.4 \pm 4.0$ Sv | $28.9 \pm 3.9$ Sv | -0.30 Sv/yr |
| ORCA025-JRA | $20.8 \pm 4.0$ Sv | $19.9 \pm 3.9$ Sv | -0.23 Sv/yr |
| ORCA025-JRA-strong | $23.9 \pm 4.1$ Sv | $22.5 \pm 4.1$ Sv | -0.31 Sv/yr |
| VIKING20X-JRA-OMIP | $34.2 \pm 4.5$ Sv | $32.2 \pm 4.6$ Sv | -0.38 Sv/yr |
| ORCA025-JRA-OMIP | $25.5 \pm 4.3$ Sv | $23.6 \pm 4.4$ Sv | -0.36 Sv/yr |
| Observations | $30.6 \pm 3.8$ Sv | $29.6 \pm 4.4$ Sv | -0.27 Sv/yr |
| | | *(2000-2009) | *(1998-2009) |

more barotropic than found in observations. Though, one finds a clear improvement of the representation of the spatial scales and location of the DWBC at 53°N in the VIKING20X-JRA-short. The density structures (grey lines in Fig. 11a-c) reveal a discrepancy between the models and observations, whereby VIKING20X better compares to the observations than ORCA025. For the comparison of NADW transports at 53°N the water mass boundary between the upper AMOC component and the

NADW was adjusted for each individual model using the mean density of the AMOC maximum at the OSNAP section over the full model integration (Handmann et al., 2018). Owing to the improved representation of lower NADW, the VIKING20X experiments simulate simulate NADW transports that fall into the observed standard deviation (Table 5). In respect to the longer-term temporal variability (Fig. 11d), it is apparent that the observations feature stronger multi-annual variability than the model experiments. There is no significant correlation between the simulations and the observed transports on interannual

timescales (here not shown). However, both observations and model experiments show a significant downward trend that is usually stronger in most of the simulations (Table 5). The general temporal evolution, though subject to much less variability on multi-annual timescales, reflects that of the AMOC (Fig. 11e). It is important to note that the AMOC in depth coordinates is of little use in the subpolar North Atlantic. In contrast to the lower latitudes, the 'overturning' is not given by an upper/warm and deeper/cold contrast but rather a strong east-west gradient responsible for the transport across density surfaces (Danabasoglu

et al., 2014; Biastoch et al., 2008a)

## 4.2 Subtropical North Atlantic

Current structures associated with the export of deep water masses from the subpolar into the subtropical North Atlantic follow interior pathways and only a specific narrow DWBC (Fig. 7). This is in particular visible around Flemish Cap and through Flemish Pass (Fig. 12). While completely absent in ORCA025, VIKING20X shows a continuous path around Flemish Cap,

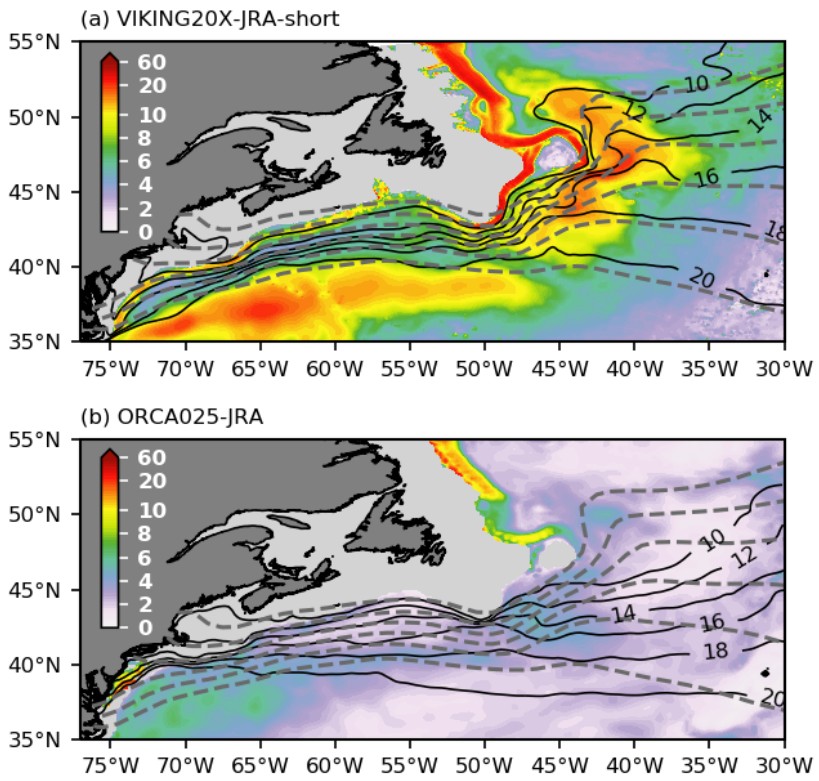

**Figure 12.** Path of the DWBC (speed on density range, see Fig. 7) and SST in degrees Celsius (solid contours) for (a) VIKING20X-JRA-short and (b) ORCA025-JRA. SST from the HadiSST dataset (Rayner et al., 2003) is shown by dashed contours.

followed by a narrow and weak current along the American shelf. Solodoch et al. (2020) noticed the strong fluctuations in the DWBC at Flemish Cap due to steep bathymetric variations. This is not necessarily an eddying signal but could also be caused by topographic Rossby waves related to Gulf Stream rings and meanders (Peña-Molino et al., 2012). Details of the deep pathways seem to be connected to the overlying flowing Gulf Stream, so that its correct separation at Cape Hatteras and path into the Northwest Corner might play an important role. Horizontal resolution is also of relevance here: Chassignet and Xu

(2017) noticed a much more realistic separation and path of the Gulf Stream if simulated at 1/25° compared to 1/12°. However, they also noted that 1/50° is required to fully represent the Gulf Stream penetration and the associated recirculation gyres at depth. Nevertheless, Figure 12 shows that the separation of the Gulf Stream and the swing into a Northwest Corner is already well reproduced in VIKING20X as indicated by the sea surface temperature (SST) distribution. In contrast, ORCA025-JRA




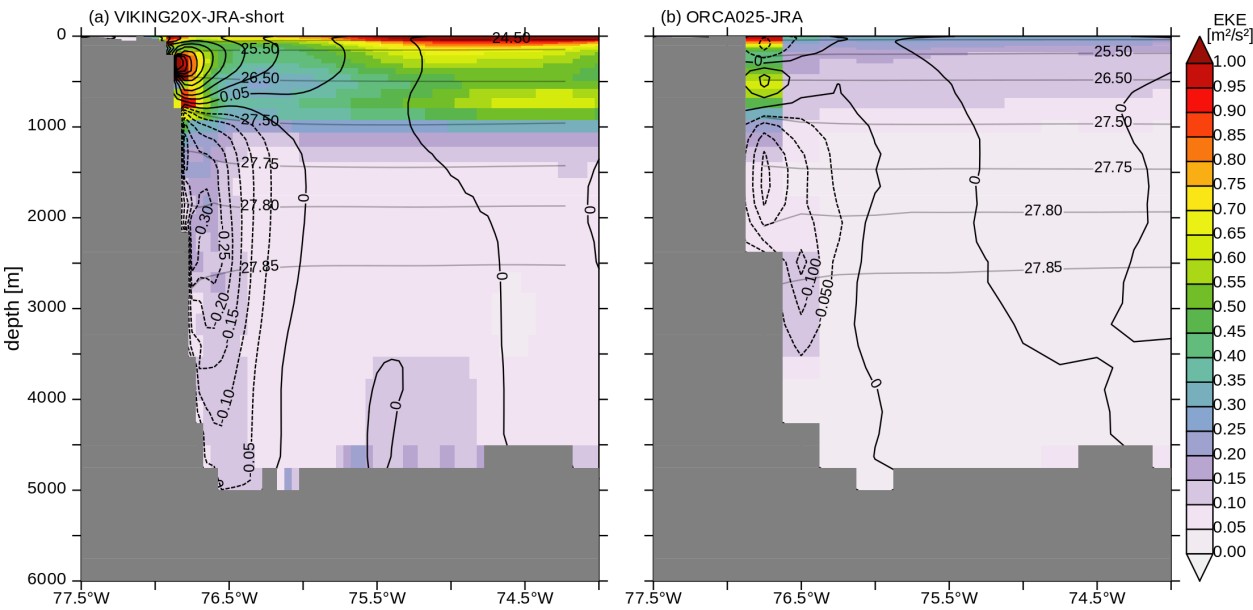

**Figure 13.** Mean (1990-2009) EKE in $m^2$ $s^{-2}$ (shaded) and meridional velocity (black contours) section at 26.5°N for (a) VIKING20X-JRA-short and (b) ORCA025-JRA with $\sigma_0$ isolines (grey).

features the too zonal path of the North Atlantic Current common for lower resolution models. This is a typical behaviour even

at 1/12°resolution as demonstrated by Chassignet and Xu (2017).

     In the subtropical North Atlantic, the southward flow of NADW is again concentrated along the western boundary (Fig. 7). It aligns with the northward flowing surface branch of the Antilles Current (Fig. 13), which is complemented by the flow through the Florida Strait. The characteristics of the AMOC introduced in the subpolar and subarctic regions, but also details of the bathymetry south of it, have an imprint in the current structure at the western boundary. In ORCA025 the flow of NADW clearly

lacks the denser part of the NADW because of the inability to maintain the overflow at this resolution, while in VIKING20X the DWBC reaches much deeper. This is also reflected in the integral measures of the AMOC (Figs. 3 and 4). The surface branch instead depends on the representation of the Bahamas Islands and the Bahamas Bank. In VIKING20X the Antilles Current is variable, eddies are crossing the section at 26.5°N northwestward, providing a prolonged maximum of eddy kinetic energy (EKE) (Fig. 13a). ORCA025 has a much weaker and stable surface transport, with even southward transport directly at

the surface.

     An important component of the RAPID observational array (and motivation for the choice of its particular latitude) are the long-term measurements of the transport through the Florida Strait obtained from voltage differences with telephone cables (e.g., Meinen et al., 2010). Table 6 demonstrates that Florida Current transports agrees well with the observations within 1-2.6 Sv for VIKING20X-CORE, VIKING20X-JRA-short and VIKING20X-JRA-OMIP. The transport is weaker in




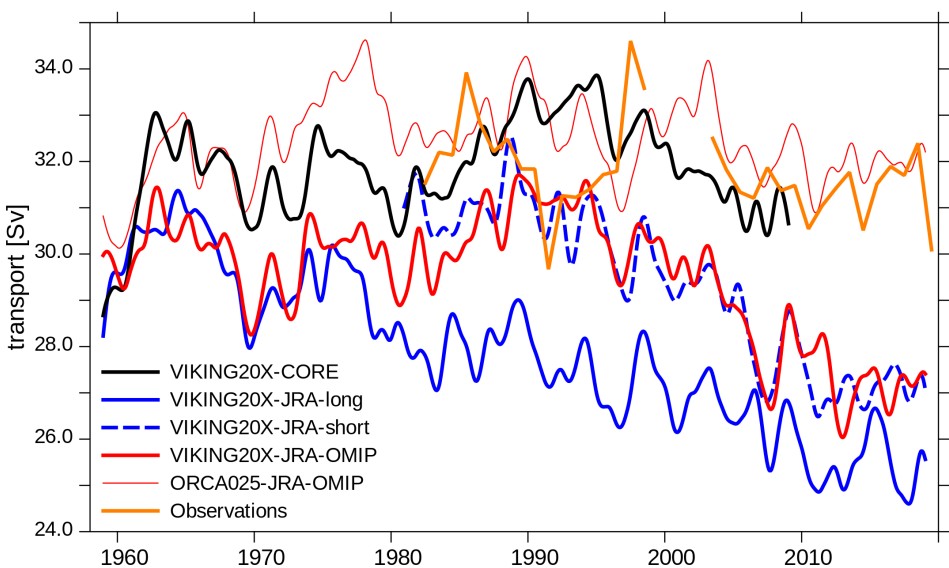

**Figure 14.** Time series of the transport through Florida Strait from experiments (interannually filtered) and from cable measurements (orange) given as yearly averages for years with data coverage > 70%.

**Table 6.** Mean and standard deviation based on monthly averaged as well as interannually filtered data of the transport through Florida Strait for the period 1990-2009. In this period, data coverage from cable measurements is 82%.

| Experiment | Mean 1990-2009 | Std.dev (mon) | Std.dev (ia) |
|---|---|---|---|
| VIKING20X-CORE | 32.1 | 1.94 | 0.98 |
| VIKING20X-JRA-short | 29.5 | 1.95 | 1.17 |
| VIKING20X-JRA-long | 27.0 | 1.55 | 0.67 |
| ORCA025-JRA | 28.9 | 2.03 | 1.12 |
| ORCA025-JRA-strong | 31.0 | 1.79 | 0.73 |
| VIKING20X-JRA-OMIP | 29.7 | 1.91 | 1.27 |
| ORCA025-JRA-OMIP | 32.6 | 1.92 | 0.75 |
| ORCA025-JRA-OMIP-2nd | 31.4 | 1.93 | 0.92 |
| Observations | 31.9 | 2.52 | 0.95 |

VIKING20X-JRA-long, which could correspond to the lower AMOC. Since wind forcing is similar in VIKING20X-JRA-short and VIKING20X-JRA-long, this difference can be attributed to the thermohaline part of the Florida Current. Variability of the Florida Current transport in all model simulations is underrepresented at monthly timescales, but comparable to observations at interannual timescales. Some of the JRA55-do experiments are correlated on interannual timescales, but we find no


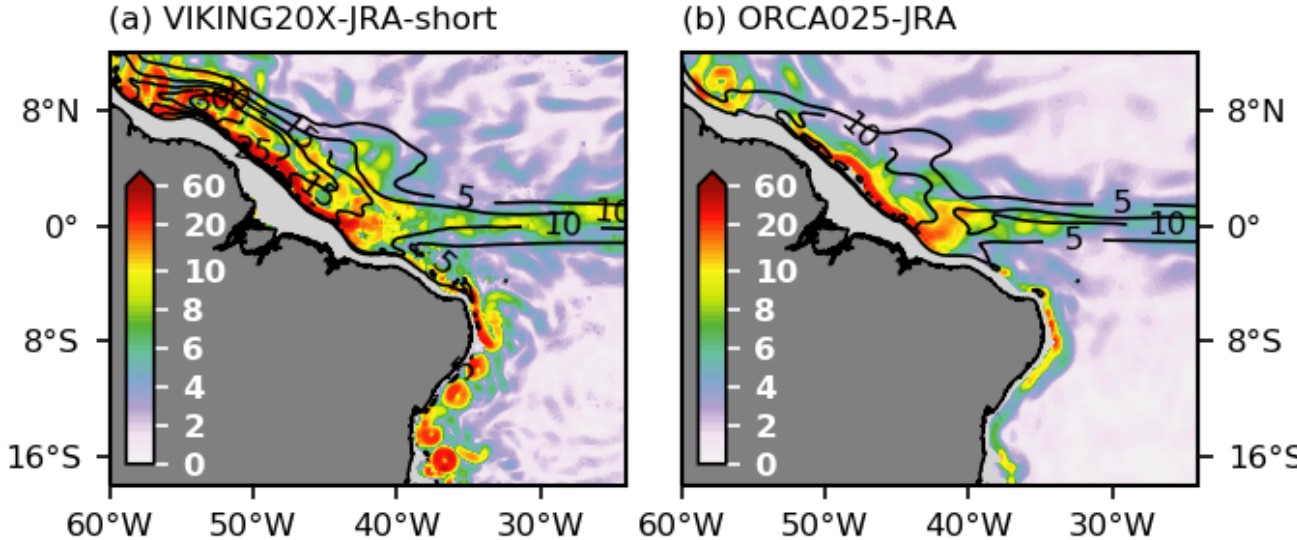

**Figure 15.** Snapshot of the path of the DWBC speed (5-day mean at date 22-02-1990, density range as in Fig. 7) and upper-ocean EKE (contours, in J m$^{-3}$, mean between 1990-2009 for the upper 1000 m) for (a) VIKING20X-JRA-short and (b) ORCA025-JRA.

correlation with the observations. Also, some experiments (e.g. VIKING20X-CORE and VIKING20X-JRA-OMIP) showed a
correlation of ∼0.75 between the Florida Current and the AMOC, while others (in particular VIKING20X-JRA-long, but also
the experiments in ORCA025) did not. This demonstrates that the Florida Current is not just a WBC closure in form of net
(wind-related) Sverdrup changes from the interior but is rather regionally influenced from the flow through the Caribbean Sea
and through regional atmospheric forcing (Lee and Williams, 1988; DiNezio et al., 2009; Hirschi et al., 2019).

Surprisingly, all VIKING20X experiments (including VIKING20X-CORE) show a declining trend that starts in the 1990s
which is not reflected in the latter part of the observations. The VIKING20X experiments forced by JRA55-do simulated a
decline of the Florida Current of ∼0.1 Sv per year over the RAPID period 2004-2018, which is about 27-43% of the AMOC
in the same period (here not shown).

### 4.3   Tropical Atlantic

From the subtropical North Atlantic towards the tropics, the NADW transport is concentrated along the western boundary (Fig.
7). Between 10°N and 10°S, offshore recirculation patterns appear more prominently than at other latitudes. Figure 15 (similar
to Fig. 7 but here as a snapshot) indicates a connection with North Brazil Current rings (Kirchner et al., 2009) that are spun
off from the reflection of the North Brazil Undercurrent (NBUC) and drift northwestward, as suggested by eddy kinetic energy





**Table 7.** Mean and monthly std. dev. of NBUC and DWBC transports across 11°S from 12 observational sections, mooring measurements and experiments for the period 2000 to 2019 (for VIKING20X-CORE 2000-2009) and sampled from monthly data according to the ship based observational coverage. The transport calculation is modified from Hummels et al. (2015) by changing from $\gamma_n$ to almost equivalent $\sigma_0$ criteria. For observations, transports based on $\gamma_n$ criteria are given as reference. (NBUC transport is calculated by integration of positive velocities in three boxes: west of 35.4°W above $\sigma_0 = 27.53$, between 35.0 and 34.65°W above $\sigma_0 = 26.73$, between 35.4 and 35°W above the line connecting $\sigma_0 = 27.53$ at 34.5° and $\sigma_0 = 26.73$ at 34.65°W; DWBC transport is calculated by integration of negative velocities west of 34.65°W between $\sigma_0$ 27.53 and second crossing of 27.88; all $\sigma_0$ criteria are based on temporally varying fields; the latter condition cannot always be fulfilled, if this second crossing does not exist, the integration is down to the bottom; these boxes are outlined in Figure 16d.)

| Experiment | NBUC+ | | DWBC- | |
|---|---|---|---|---|
| | (2000-2019) | (obs) | (2000-2019) | (obs) |
| VIKING20X-CORE | $33.5 \pm 5.7$ | | $33.6 \pm 15.2$ | |
| VIKING20X-JRA-short | $27.8 \pm 4.1$ | 28.7 | $31.1 \pm 12.6$ | 33.7 |
| VIKING20X-JRA-long | $23.6 \pm 3.8$ | 23.4 | $25.6 \pm 10.4$ | 31.6 |
| ORCA025-JRA | $18.8 \pm 3.8$ | 19.3 | $21.7 \pm 5.7$ | 22.6 |
| ORCA025-JRA-strong | $21.4 \pm 3.6$ | 22.0 | $24.8 \pm 6.0$ | 27.4 |
| VIKING20X-JRA-OMIP | $29.4 \pm 4.3$ | 30.5 | $32.0 \pm 12.6$ | 38.1 |
| ORCA025-JRA-OMIP | $24.8 \pm 4.0$ | 25.1 | $28.1 \pm 6.1$ | 30.4 |
| ORCA025-JRA-OMIP-2nd | $22.5 \pm 3.8$ | 23.7 | $26.6 \pm 6.0$ | 29.5 |
| Moorings $(\gamma_n)^*$ | $25.9 \pm 4.5$ | | $20.0 \pm 9.2$ | |
| Ship sections $(\gamma_n)$ | | 25.5 | | 30.4 |
| Ship sections $(\sigma_0)$ | | 25.4 | | 30.4 |

$^*$(interrupted from 2005-2012)

(EKE) much better represented in VIKING20X-JRA-short than ORCA025-JRA. Schulzki et al. (manuscript under review at *J. Geophys. Res.*) describe that recirculation pattern and instabilities in the DWBC can be related to incoming Rossby waves and

mesoscale eddies. Further south, at around 6-8°S, the DWBC splits into eddies transporting the deep water in their cores to the south, consistent with observations (Dengler et al., 2004). Due to the coherent pathways, the time-mean field (Fig. 7) shows a continuous path, but also displays the enhanced values of the offshore recirculations described above. The eddy pathway merges again as a DWBC, then fades out at the Vitória-Trindade Seamount Chain at around 20°S.

Measurements and long-term monitoring of the WBC system off Brazil have been motivated by the concentration of the
northward upper-ocean flow in the NBUC (Hummels et al., 2015). They are part of the 'Tropical Atlantic Circulation and Overturning at 11°S' (TRACOS) array (Herrford et al., 2021), which consists of bottom pressure observations at 300 m and 500 m depth since 2013 on both sides of the basin in order to obtain an AMOC estimate, the mentioned long-term western boundary array (since 2000, but with a gap between 2004 and 2013) and current observations at the eastern boundary off Angola since 2013. Rühs et al. (2015) show that the NBUC transport can be used as an indicator for the upper branch of



**Figure 16.** Mean along-shore velocity (m s$^{-1}$) section at 11°S for (a) ORCA025-JRA, (b) VIKING20X-CORE, (c) VIKING20X-JRA-short and (d) from 12 observational ship sections in the periods 2000-2004 and 2013-2019 with $\sigma_0$ isolines. Dashed boxes in (d) indicate the area taken into account for NBUC and DWBC transport calculations.

the AMOC if the horizontal wind-driven circulation is also accounted for. In addition to the northward return flow associated with the AMOC, the NBUC also carries most of the equatorward flow related to the South Atlantic Subtropical Cell (Schott et al., 2004). Figure 16 shows that the general structure of the WBC system is visible in all experiments. The NBUC with its





subsurface maximum and the DWBC below already exist in ORCA025-JRA. The eddy-rich configurations better represent the elongated subsurface core of the NBUC and the wider, eddying (cf. Fig. 15a) DWBC, merging with a recirculation pattern

offshore of the (deeper part of the) NBUC. The representation of the water masses (here indicated by the density lines) appears quite well. Model transports are usually within the standard deviations of the observational estimates based on the moorings, but are generally too weak at low resolution and by 1/3 (NBUC) and 2/3 (DWBC) too strong for VIKING20X-CORE (Table 7).

It is interesting to note and a guidance for future model-observation comparisons that even for a (with 12 ship-based ob-

servations) well-covered section a detailed temporal selection of model output can be important depending on the variability of the system in question (Schwarzkopf, 2016). While the values deducted from the moorings show a good agreement with the ship sections for the NBUC, those for the DWBC are off by more than 10 Sv (Table 7). This is due to the fact that the ship-based estimate is biased by intra-seasonal variability (Hummels et al., 2015), with ship sections often conducted during times of maximum southward flow that often only lasts for a few days (Fig. 17e, especially during 2000-2004). It is important

to emphasise that the simulations capture the strong variability and confirm the apparent discrepancy between the long-term and the subsampled values.

Figure 17a shows the temporal evolution of the AMOC at 11°S. In contrast to the one at 26.5°N (Fig. 5a), it shows a minimum in the late 1960s which is in particular the case in VIKING20X-JRA-OMIP that started one decade earlier from rest. In the following decades, the experiments (apart from VIKING20X-JRA-long) simulate an increase into the 1990s with a

decline thereafter.

Transports of the NBUC and DWBC are less robust among the experiments and show no clear or consistent multi-decadal evolution. In consequence, trends over limited periods are less consistent. Apart from the trends, both AMOC and NBUC are significantly correlated between individual experiments on interannual timescales with $r$ values of up to 0.85. In contrast, DWBC transports are rarely correlated.

Within individual experiments, there is no robust co-variability of the AMOC with NBUC (or DWBC) transports. Some (e.g. VIKING20X-JRA-OMIP) show significant correlations while others (e.g. VIKING20X-JRA-long) do not (here not shown).

### 4.4   Subtropical South Atlantic

Entering the subtropical South Atlantic brings us closer to the southern boundary of VIKING20X's high-resolution nest. In the following we explore the ability of the nested configuration to simulate the mesoscale circulation in the Agulhas Current system

and the Brazil-Malvinas confluence and if the host model is capable in correctly simulating the transports at the SAMOC observations which are placed just outside the nested area. For comparison we use INALT20 (Schwarzkopf et al., 2019), a configuration that is in large parts (in particular resolution and atmospheric forcing) similar to VIKING20X, but with an eddy-rich nest reaching into the Southern Ocean and into the western Indian Ocean.

Figure 18 confirms the general ability of VIKING20X to simulate parts of the mesoscale in the vicinity of the southern

nest boundary. It is logical that the variability in the Brazil-Malvinas confluence in VIKING20X-JRA-short is lower and rather comparable to ORCA025-JRA (Fig. 18a and b), given the fact that this latitudes are only represented on the host grid at




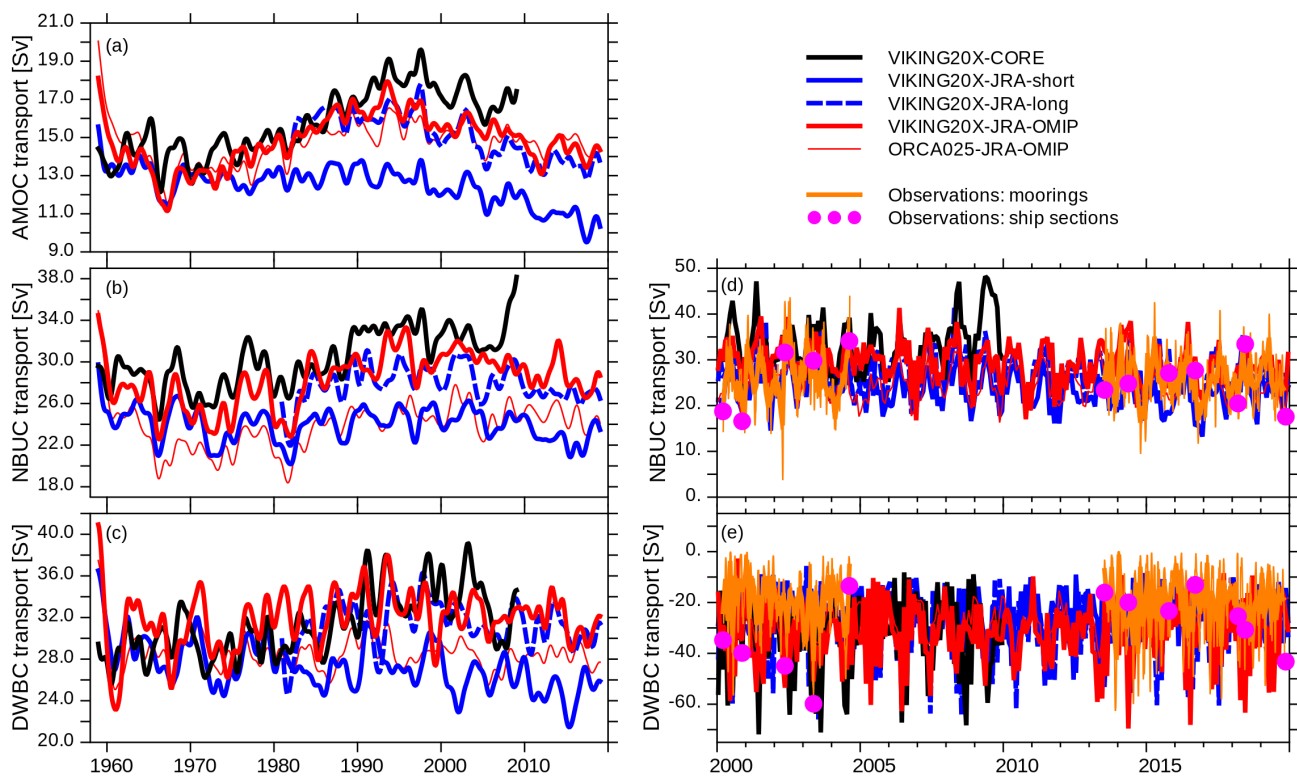

**Figure 17.** Time series of interannually filtered (a) AMOC, (b) NBUC and (c) DWBC transports at $11°$S; monthly averages of positive (NNE-ward) NBUC and negative (SSW-ward) DWBC transports (see Table 7 for details of the definitions) are given in (d) and (e) together with mooring based (orange curves; thick monthly, thin 2.5-day averages) and ship based observations (purple dots)

$1/4°$resolution. In addition to the mesoscale signal Schwarzkopf et al. (2019) demonstrated that the correct representation of the confluence region has consequences for the structure and transport of the Malvinas Current. The picture is similar for the path of Agulhas rings. Here, the formation process in the retroflection of the Agulhas Current south of Africa is outside the

VIKING20X nest. Since Agulhas rings are generally represented in ORCA025 (Schwarzkopf et al., 2019), they also enter the nest. The difference compared to a configuration fully resolving the Agulhas Current dynamics (and observations) is that ORCA025 simulates too regular Agulhas rings. This results in too regular pathways into the South Atlantic (cf. Fig. 18b and d).

The AMOC at $34.5°$S shows the same evolution as the one in the North Atlantic, with a maximum in 1990s (in some

experiments relative and generally less pronounced) and a weakening thereafter if performed under JRA55-do forcing (Fig. 19). Even though the representation of the mesoscale is different as discussed above, the mean, interannual variability and long-term trend in INALT20-JRA-long and VIKING20X-JRA-long are remarkably comparable. By comparing the transports in VIKING20X-JRA-long with those in ORCA025-JRA, and VIKING20X-JRA-OMIP with ORCA025-JRA-OMIP we notice





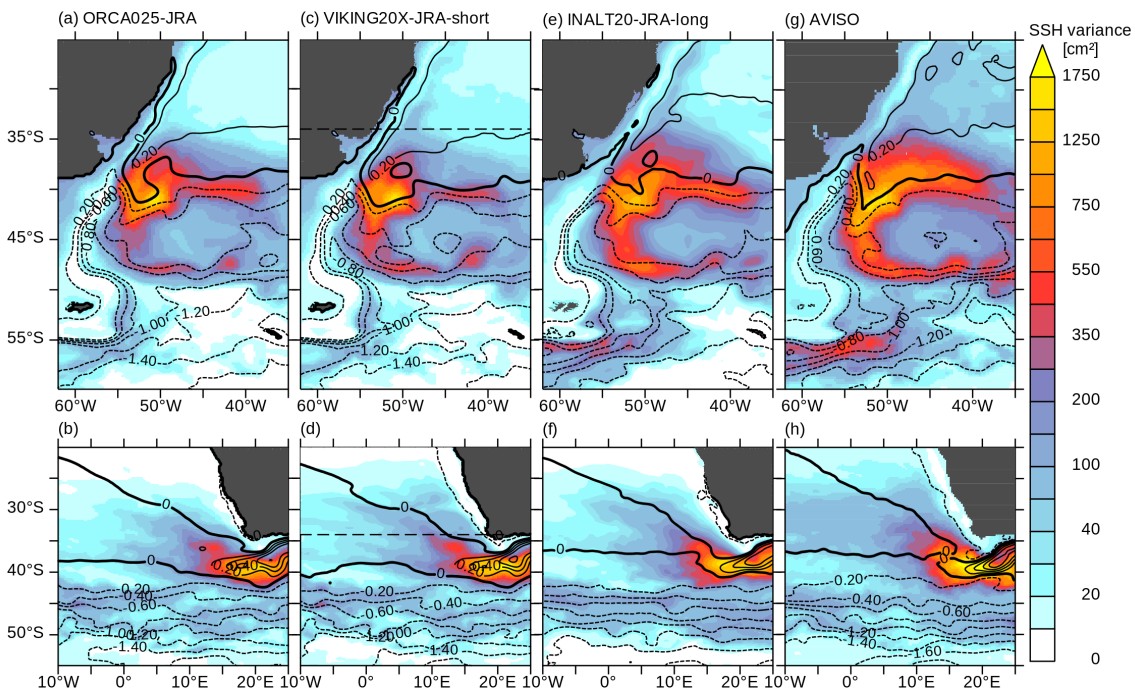

**Figure 18.** Sea surface height in the Brazil-Malvinas Confluence zone (upper row) and west Agulhas region (lower): 1990-2009 mean (contours, in m) and variance (shaded) in (a,b) ORCA025-JRA, (c,d) VIKING20X-JRA-short, (e,f) INALT20-JRA-long and (g,h) from satellite altimetry. Dashed lines in (c,d) mark the southern boundary of the nested region in VIKING20X.

a similar resolution effect as seen in the North Atlantic (Table 8). Compared to the (temporally limited) observations by Meinen
et al. (2018), most experiments underestimate the mean AMOC transport through the section. However, all VIKING20X
experiments are in the range of the standard deviations in respect to the monthly variability. The representation of Agulhas
rings even in the eddy-present ORCA025 resolution could be an important prerequisite for the realistic AMOC variability.
The interannual variability is correlated among experiments, with $r$-values of, e.g., 0.8 between VIKING20X-JRA-long and
VIKING20X-JRA-OMIP, similar to the other latitudes (here not shown). Interestingly, this is also the case when mesoscale
variability is better resolved. INALT20-JRA-long interannually correlates with VIKING20X-JRA-OMIP at $r=0.81$. Given the
fact that the generation of Agulhas rings is a highly stochastic process (Biastoch et al., 2009), this shows that the interannual
variability of the AMOC at 34.5°S has a significant deterministic component by the atmospheric forcing.

## 5   Discussion and conclusions

Our results show that a 'realistically' configured ocean hindcast configuration like VIKING20X is well suited for simulating
the large-scale circulation dynamics in the Atlantic Ocean. Depending on a proper representation of the mesoscale provided





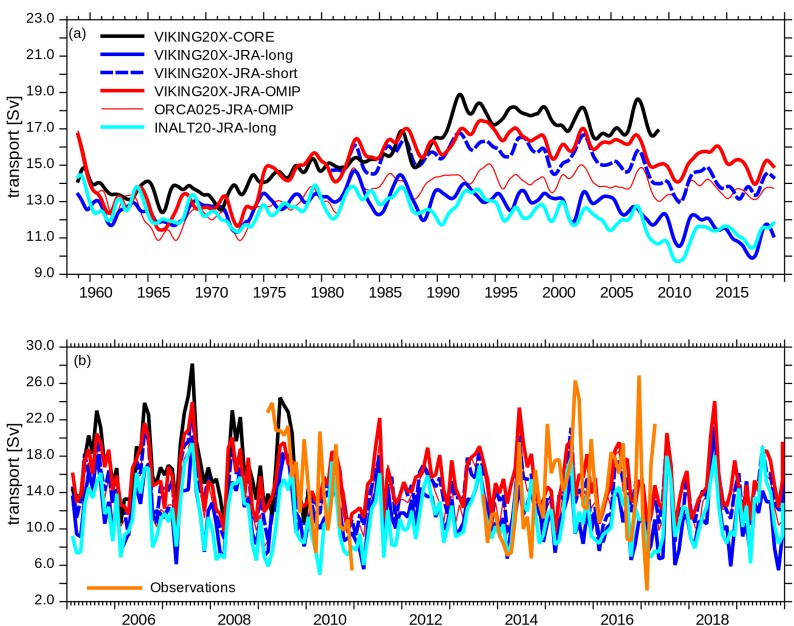

**Figure 19.** Time series of the AMOC strength at 34.5°S (a) interannually filtered and (b) monthly averages for the most recent 15 years with observations from Meinen et al. (2018) (orange).

**Table 8.** Mean and standard deviation based on monthly data of the AMOC transport at 34.5°S for the period 1990-2009 (for VIKING20X-CORE only the overlapping year 2009 is used. *The observational period covers the years 2009/2010 and 2013-2017).

| Experiment | 1990-2009 | obs* |
|---|---|---|
| VIKING20X-CORE | 17.5 ± 3.6 | 17.2 ± 5.2 |
| VIKING20X-JRA-short | 15.5 ± 3.3 | 13.6 ± 2.9 |
| VIKING20X-JRA-long | 13.0 ± 3.2 | 11.1 ± 2.5 |
| ORCA025-JRA | 10.7 ± 3.2 | 8.6 ± 2.7 |
| ORCA025-JRA-strong | 11.7 ± 3.2 | 10.5 ± 2.7 |
| VIKING20X-JRA-OMIP | 16.3 ± 3.1 | 15.0 ± 2.8 |
| ORCA025-JRA-OMIP | 14.2 ± 3.2 | 13.3 ± 2.8 |
| ORCA025-JRA-OMIP-2nd | 12.3 ± 3.2 | 11.0 ± 2.9 |
| INALT20-JRA-long | 12.3 ± 3.2 | 10.8 ± 3.0 |
| Observations | | 14.7 ± 5.4 |



by an adequate resolution over the full model domain and the availability of an accurate atmospheric forcing, many aspects of the simulated wind-driven and thermohaline circulation compare very well with observations. These include the large-scale structure of the mean flow, the distribution and strength of mesoscale eddies, WBC structures and individual current systems. There is good agreement between model and observations in terms of velocity structures and integral transports, even of their

temporal variability and trends. Because of the strong impact of the wind forcing, the role of the atmospheric forcing data set cannot be underestimated for such an ocean-only model. With the shift of the ocean model community from CORE forcing (Large and Yeager, 2009) to the higher resolved and updated JRA55-do (Tsujino et al., 2018) forcing comes an important change. CORE was known to enhance individual wind systems such as the equatorial trades or the Southern Hemisphere westerlies (Large and Yeager, 2009; Brodeau et al., 2010) which could be the cause for generally higher WBC transports in

VIKING20X-CORE. Together with a different subpolar/subarctic freshwater budget, e.g. by the precipitation components and river input, this leads to a stronger and more stable AMOC, as well as its regional components.

Even though WBC transports and AMOC strength are generally enhanced at eddy-rich resolution, other aspects are remarkably independent. One example is the decadal transport variability of the subpolar gyre. It is important to note that our configurations do not allow to isolate the effect of eddies. We acknowledge that the eddy-present ORCA025 does not use any

eddy-parameterisation, it explicitly simulates 'some' (the larger-scale) part of the mesoscale spectrum, while neglecting others. This dilemma is similar to many modern eddy-present configurations performed for CMIP6 (Hewitt et al., 2020). And yet, although the level of mesoscale variability significantly increases from $1/4°$ to $1/20°$, the interannual variability of the AMOC appears quite robust. Features like the 2010 minimum can be simulated well at eddy-present resolution, pointing to an ability of the CMIP6 suite in simulating an important part of the mesoscale contribution to the AMOC.

More caution is required with respect to the thermohaline driven part of the circulation. Although we see an improvement in the deep convection regions in the Labrador and Irminger Seas from $1/4°$ to $1/20°$, the amount of potentially produced upper NADW (here quantified in terms of the MLD volume) appears remarkably robust. In contrast, for the backbone of the AMOC, the overflow across the Greenland-Scotland Ridge is clearly improved with enhanced horizontal resolution. And still, the underestimation of the lower NADW at 53°N confirms that the entrainment of ambient water masses into the overflow

along its downward descent while circling the subpolar gyre even simulated at high resolution is subject to spurious mixing. The simulations cannot maintain the observed high densities to yield the correct transports in the densest levels. The underrepresentation of the lower NADW throughout the whole Atlantic still remains an important challenge in z-coordinate models (Legg et al., 2006), even though an eddy-rich resolution is an important improvement. As a cautionary note, we point to the study by Colombo et al. (2020) who showed that a higher vertical resolution does not automatically enhance the spreading of

dense overflows and sometimes even leads to the contrary effect.

For the estimation of the decadal variability and any long-term trend of the AMOC, realistically distributed components of the freshwater budget are crucial because of the enhanced sensitivity of ocean-only models to freshwater fluxes, in particular their corresponding subpolar freshwater budget (Griffies et al., 2009; Behrens et al., 2013). Because of this sensitivity, AMOC trends from internal variability and external forcing are usually difficult to quantify against model-related drifts. The modelled

trends depend on artificial choices of the prescribed freshwater fluxes, such as the required strength of the SSS restoring or (as





in our case) a global balance of the freshwater budget. Since these parameter or forcing choices are not well constrained and may even interfere with each other, the quantification of the AMOC evolution over the past decades from hindcast simulations remains a challenge . Any quantification of global warming related trends induced by the atmospheric forcing is compromised by the underlying model drift.

Beyond the description and verification of the VIKING20X set of experiments, our main objective is to test whether regional current systems are able to detect AMOC changes. In the remaining discussion, we address this task by answering three questions: (1) How coherent are AMOC changes across latitudes throughout the Atlantic Ocean? (2) Are AMOC trends detectable in regional current systems? And if so, (3) can regional observations help to verify and confirm modelled AMOC trends?

*(1) How coherent are AMOC changes across latitudes throughout the Atlantic Ocean?*

Most VIKING20X experiments (apart from VIKING20X-JRA-long showing a continuous decline of the AMOC throughout the whole hindcast period) show a similar evolution of the AMOC, with an increase towards the mid-1980s and 1990s and a decline thereafter. This is exemplary provided for VIKING20X-JRA-OMIP (Fig. 20). The Hovmoeller diagram (here given in density coordinates to correctly address the subpolar North Atlantic) shows the evolution and the spreading of AMOC anomalies throughout the Atlantic Ocean. Consistent with earlier findings, the strongest anomalies arise in the subpolar gyre

as a response to a decadally varying heat and freshwater forcing (Biastoch et al., 2008a; Yeager and Danabasoglu, 2014). On decadal timescales, this signal is evident at all latitudes, but with decreasing amplitudes towards the south (cf. Figs. 20a and c, but also Figs. 5a, 17a, and 19a). Other signals such as the minimum in the late 1960s or the maximum in the 2000s seem to be of southern origin and remain restricted until ∼20°N. AMOC anomalies appearing on shorter timescales add interannual 'noise' to this decadal evolution. We identify both latitudinally restricted changes of interannual variability, but also anomalies

propagating from the north or the south throughout the full Atlantic Ocean. These anomalies are either caused by local wind changes inside the high-resolution nest itself or provided from the eddy-present host grid outside of the nested domain, which rapidly propagate north- and southward via topographic and shelf waves along both sides of the Atlantic Ocean (Getzlaff et al., 2006; Biastoch et al., 2008b). In the Hovmoeller diagram the associated patterns are only slightly slanted, indicating the fast wave propagation. Only Besides the typical equatorial discontinuity, anomaly propagation is significantly delayed at the

transition between the subpolar and the subtropical North Atlantic because of the complex exchange between the gyres (Zou et al., 2019).

If we now focus on the 'trend' of the AMOC over the recent two decades (Fig. 21), which itself is part of a multi-decadal evolution, VIKING20X-JRA-OMIP shows a coherent spin-down of the NADW cell. The general evolution of the other experiments is similar but differs in the strength of the decline (Fig. A2). Since all experiments are based on the same atmospheric

forcing, differences in the decline are due to choices of the freshwater application and the initialisation. Changes in the overflow and subpolar deepwater formation provide the main origin for the AMOC evolution, and consequently the decline (once provided in density coordinates to correctly address the subpolar North Atlantic, Fig. 21b) is strongest north of ∼40-45°N. Quantitatively, the AMOC decline peaks at ∼60°N, the latitude close to the southern tip of Greenland until which the overflow has entrained most of its ambient water masses (Fig. 21c). South of the transition towards the subtropical North Atlantic,

at around 35°N, the decline in VIKING20X-JRA-long and VIKING20X-JRA-OMIP is consistent 1-1.5 Sv per decade, even



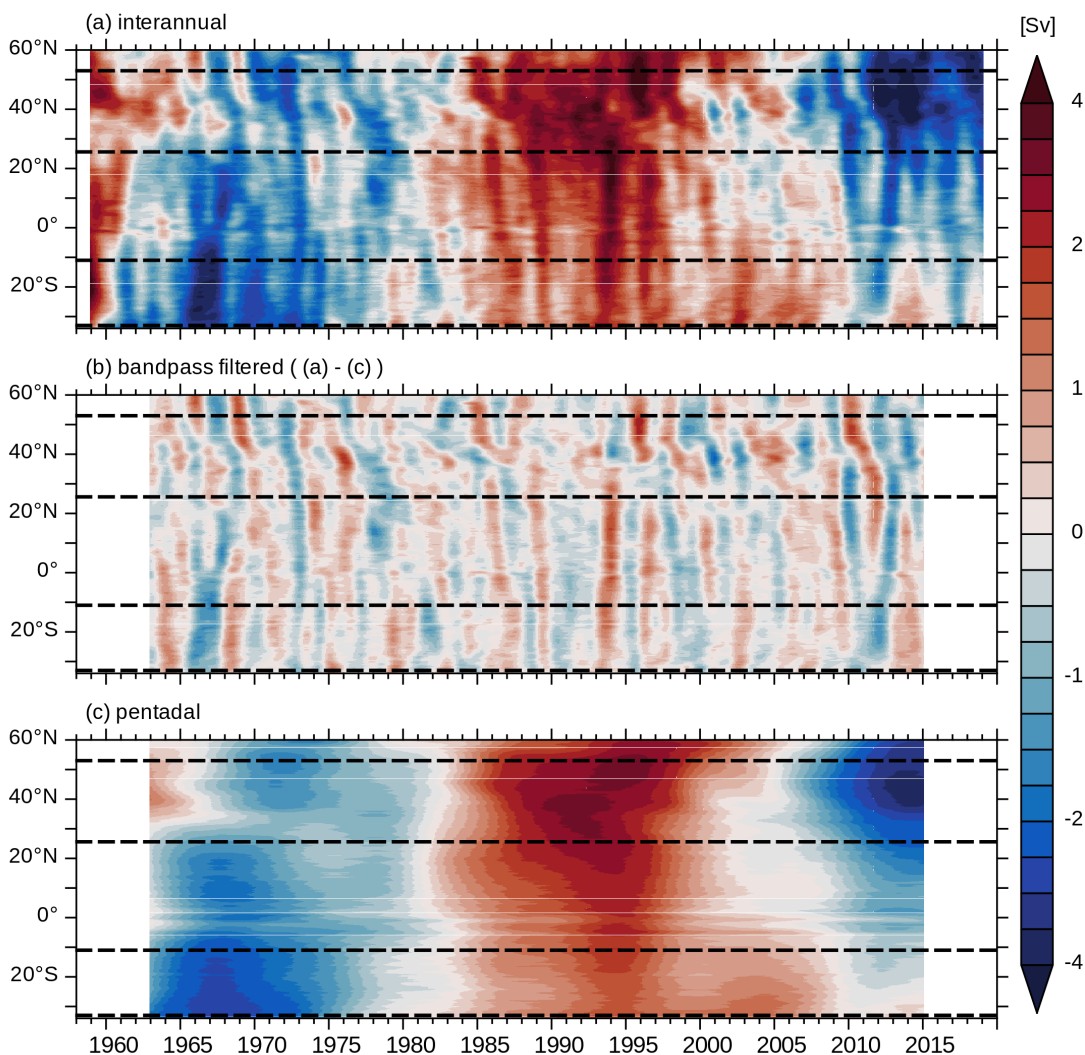

**Figure 20.** Hovmoeller diagrams of AMOC anomalies in density coordinates given as (a) interannually (b) bandpass and (c) pentadally filtered time series of the maximum strength between $\sigma_2=36$ and 37 from VIKING20X-JRA-OMIP. Lines indicate latitudes considered in this study.

though their mean AMOC strength differs by more than 4 Sv. (Fig. 21c, see also Table 2). VIKING20X-JRA-short features a stronger decline since it starts from a relatively high value as a result of the preceding CORE forcing and obviously experiences a spindown from the high level of the CORE forcing. Owing to the general spindown of the NADW cells we conclude

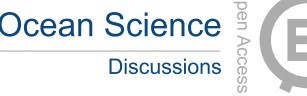

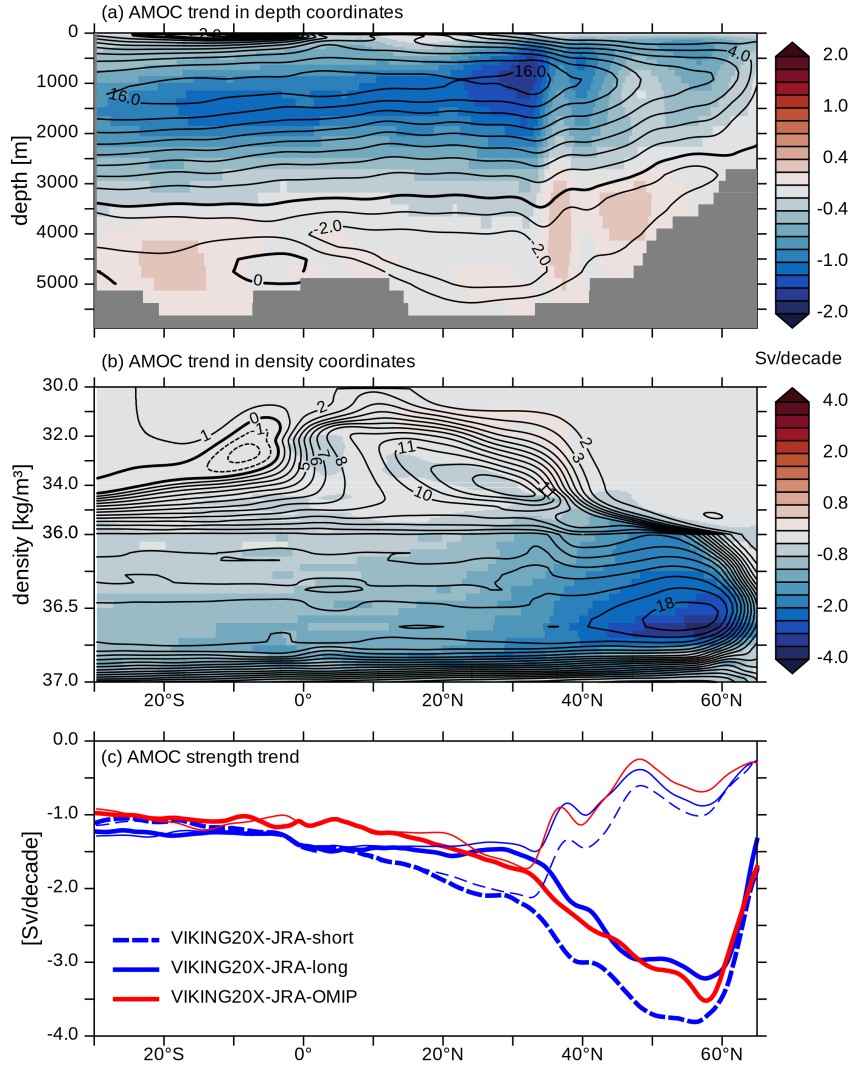

**Figure 21.** Mean (contours) and linear trend (shading) in the AMOC for the period 2000 to 2019 in VIKING20X-JRA-OMIP (a) in depth and (b) density ($\sigma_2$) coordinates. (c) shows the linear trend of the vertical maximum in the AMOC in depth coordinates below 400 m (thin) and in density coordinates below $\sigma_2 = 36$ (thick). All fields are meridionally smoothed with a Hanning filter of $10°$ window size.

that decadal and longer-term AMOC trends could, in principle, be estimated throughout all latitudes of the North and South

Atlantic but with different strengths.

   *(2) Are AMOC trends detectable in regional current systems?*





**Table 9.** Linear trends (Sv per decade, all based on monthly averaged data) in AMOC strength (in depth or density coordinates) and western boundary current transports at different latitudes for the period 2000-2019 from the three experiments in VIKING20X under JRA55-do forcing. Observational estimates cover different periods, as indicated.

| | 53°N | | 26.5°N | | 11°S | | | 34.5°S |
|---|---|---|---|---|---|---|---|---|
| Experiment | AMOC($\sigma_2$) | DWBC | AMOC($z$) | FC | AMOC($z$) | NBUC | DWBC | AMOC($z$) |
| VIKING20X-JRA-short | -3.6 | -3.8 | -2.0 | -1.6 | -1.2 | -1.6 | -1.0 | -0.9 |
| VIKING20X-JRA-long | -3.0 | -2.6 | -1.3 | -1.0 | -1.4 | -1.4 | -0.7 | -1.3 |
| VIKING20X-JRA-OMIP | -2.9 | -2.6 | -1.5 | -1.6 | -1.1 | -2.1 | -1.1 | -0.7 |
| Observations | | -2.5 | -1.3 | -0.3 | | -0.1 | +0.8 | - |
| | | (2000-2017) | (2004-2018) | | (2000-2004 & 2013-2019) | | | |

Apart from the basin-wide estimate at 26.5°N, regional observations exist and contribute long-term observations of key components of the AMOC. In this analysis we have focused on three time series, the WBC array at 53°N monitoring DWBC transports at the exit of the Labrador Sea, the transport through the Florida Strait an important part of the upper branch of the AMOC, and the WBC array at 11°S in which both the upper and the deep branch of the AMOC are assumed to be concentrated along the western boundary.

It is intriguing that the DWBC trend at 53°N is able to capture the of the AMOC trend calculated in density coordinates within 10-15% (Table 9). This correspondence even holds for the stronger trends in VIKING20X-JRA-short. This agreement confirms the high potential of WBC measurements at 53 °N (Fischer et al., 2004; Zantopp et al., 2017; Handmann et al., 2018) to truly monitor changes of the AMOC in the subpolar North Atlantic (Böning et al., 2006).

For the transport through the Florida Strait and its agreement with AMOC changes at 26.5°N, we get a different picture: In VIKING20X-JRA-short and VIKING20X-JRA-long, the decline of the Florida Current is about 75-80% of the decline of the AMOC; in contrast to VIKING20X-JRA-OMIP where it is stronger. This may indicate that the Florida Current could act as a general precursor of the AMOC trend, but is unable to exactly quantify it. This is confirmed by the observations itself, where the Florida Current decline represents just 15% of the AMOC decline. It is obvious that the Florida Current is not just a simple closure of Sverdrup dynamics but rather shielded by the shallow bathymetry west of the Bahamas and fed through the Gulf of Mexico, with upstream anomalies determining its transport (Hirschi et al., 2019). This may also explain the disagreement between modelled and observed trends of the Florida Current, even though both show comparable mean transports (Table 6.

At 11°S, the WBC system is more exposed to the open ocean. Although the experiments show a robust AMOC decline of about 15%, their NBUC changes are much less consistent and do not necessarily reflect the AMOC: for VIKING20X-JRA-long the NBUC trend does equal the one of the AMOC, while for VIKING20X-JRA-OMIP the NBUC trend is almost twice as large as the AMOC trend. For the DWBC instead, trends in VIKING20X-JRA-short and VIKING20X-JRA-OMIP fit to that of the AMOC, whereby VIKING20X-JRA-long simulates a 50% weaker trend than for the AMOC. While the signs of





NBUC and DWBC trends still point towards a spin-down of the AMOC cell, an exact quantification of the reduction remains
challenging. Apart from changes in the interior, not represented by the boundary current measurements, the disagreement can
also be attributed to the overlying wind-driven subtropical cell (viz. counter-clockwise circulation in the upper few 100 meters
in Fig. 3 and in Fig. 21b), at this latitude requiring the addition of the wind-induced gyre circulation for the interpretation
of AMOC changes (Rühs et al., 2015). We also have to acknowledge that the large variability of transports associated with
mesoscale structures and the large gaps in the observational record between 2005 and 2012 adds another line of complication
and requires longer time series to draw consistent conclusions. At 34°S, we note a disagreement of AMOC trends in the
different experiments. As shown by Fig. 20, AMOC anomalies are entering from the south and interfere with the anomalies
arriving from the north. These may have an impact on the calculation of longer trends.

*(3) Can regional observations help to verify modelled AMOC trends?*

Together with the AMOC measurements at 26.5°N, the WBC array at 53°N provides the longest observational time series.
As shown above, the latter has (according to the model) a good potential to provide trends of the basin-scale AMOC. The
observational AMOC estimate at 26.5°N indicates a decline that is in the range of all three experiments (Table 9), again with a
stronger trend in VIKING20X-JRA-short. We note that the observational AMOC trend has to be taken with caution since the
evolution of the AMOC measured at RAPID does not show a continuous decline but rather a strong minimum around 2010
and a stabilisation thereafter (Fig. 5a). A strong argument for a realistic AMOC trend in the past two decades emerges from the
comparison at 53°N. Both VIKING20X-JRA-long and VIKING20X-JRA-OMIP are within 0.1 Sv of the observed reduction of
the DWBC transport of 2.5 Sv per decade (Table 9), hence seem to realistically simulate the decline of the subpolar AMOC. For
the subtropical North Atlantic at 26.5°N this reduces 1.3 Sv, again with VIKING20X-JRA-long and VIKING20X-JRA-OMIP
realistically representing the AMOC as derived from RAPID.

Apart from the observation-model comparison, additional insight comes from the Ocean Model Intercomparison Project
2 (OMIP-2), performed under JRA55-do forcing. According to Tsujino et al. (2020), the ensemble average of 11 models,
performed with different numerical code bases and mainly configured at eddy-parameterising (few at eddy-present) resolutions,
show a linear trend of -1.19 Sv per decade over the time frame 2000-2018 at 26.5°N, which is weaker than RAPID. We cannot
conclude if the stronger decline simulated by VIKING20X-JRA-OMIP and VIKING20X-JRA-long is caused by the better
representation of mesoscale eddies, as indicated by the resolution dependence throughout this study, or by the details of the
freshwater flux application. An important factor could be the 5th cycling of the simulations through the forcing period done for
OMIP-2. While 5 cycles are not achievable at such high resolution, the comparison of the 1st and the 2nd cycle of ORCA025-
JRA-OMIP already suggests that this may play a role. The trend in ORCA025-JRA-OMIP reduces from -0.88 Sv per decade
in the 1st cycle to -0.67 in the 2nd cycle. On the other hand, it is quite foreseeable that the overall strength of the AMOC and
the related water masses may also drift away from the observations causing a general reduction of the AMOC strength and its
key components (Fig. 8).

Even though we find good agreement with estimates at 26.5°N and 53°N, we can not conclusively assess how much of the
trends simulated by the different experiments are still related to the model settings. We can also not be entirely sure whether
the trends provided by the JRA55-do forcing are realistic. If we consider VIKING20X-JRA-OMIP as the most promising





simulation given good agreement of mean transports, it is clear that the AMOC is subject to multi-decadal variability with a
stable evolution in the 1960s and 1970s, an increase towards the mid-1990s and a decline thereafter. For the past two decades
our experiments suggest that the AMOC (in density coordinates) in the subpolar North Atlantic was subject to a decline of up
to 3 Sv. For the subtropical North Atlantic and further south this reduces to about 1.5 Sv and less. This is generally in line with
a compilation of proxy observations presented by Caesar et al. (2021). On a longer timescale, it also fits the 4-Sv decline from
the 1950s/1960s towards the recent decade indicated by an SST-based proxy (Caesar et al., 2018).

What is needed to better quantify trends in future experiments and limit the influence of model drift? Apart from further
improvements of ocean model configurations, it is clear that multi-decadal hindcasts would directly benefit from a better
closure of the heat and freshwater budget. This can only be achieved up to a certain degree since the fluxes are by construction
less variable due to the prescribed atmospheric state. Further relaxation of the Bulk formulae or a move towards coupled
atmosphere-ocean models may help. While the latter are now routinely available, even at basin-scale mesoscale resolution
(Matthes et al., 2020), additional 'constraints' such as the 'partial coupling' approach described by Thoma et al. (2015), could
be a potential solution to re-introduce the interannual to multi-decadal hindcast 'timing' into coupled experiments.

In this study we had to concentrate to a limited set of long-term observations. In addition, a number of historic ocean
observations exist, from individual measurements dating back to the 19th century, repeated ship sections during the WOCE era
in the 1990s, to a drastic increase through satellite measurements and autonomous instruments such as ARGO in the 2000s.
Ocean modellers usually make use of those for model initialisation or verification. More systematic approaches to combine
model and data through assimilation are powerful, but also fail in terms of their ability to exactly quantify the required trends
(Karspeck et al., 2015; Jackson et al., 2019). A novel route in this respect that has only been started to be explored, are data
science approaches. These have demonstrated to push the limits of the interpretation of big data and provide insight not only
to pattern and distributions but also the interpretation (and ultimately the understanding) of dynamics (Sonnewald et al., 2019;
Aksamit et al., 2020; Reichstein et al., 2019). Nevertheless, our results demonstrate the value and importance of thoroughly
and carefully adjusting forcing, grid resolution and settings of 'classic' ocean models to the tasks of simulating the AMOC and
filling observational gaps for the benefit of an improved understanding of the ocean.





*Code and data availability.* The NEMO code is available at https://www.nemo-ocean.eu. For reproducibility of all results, the scripts as well as all data required to produce the figures are made available through GEOMAR (https://hdl.handle.net/20.500.12085/a8f98a1a-473f-11ea-a036-c81f66eb46c3). Additional model output is provided on request.

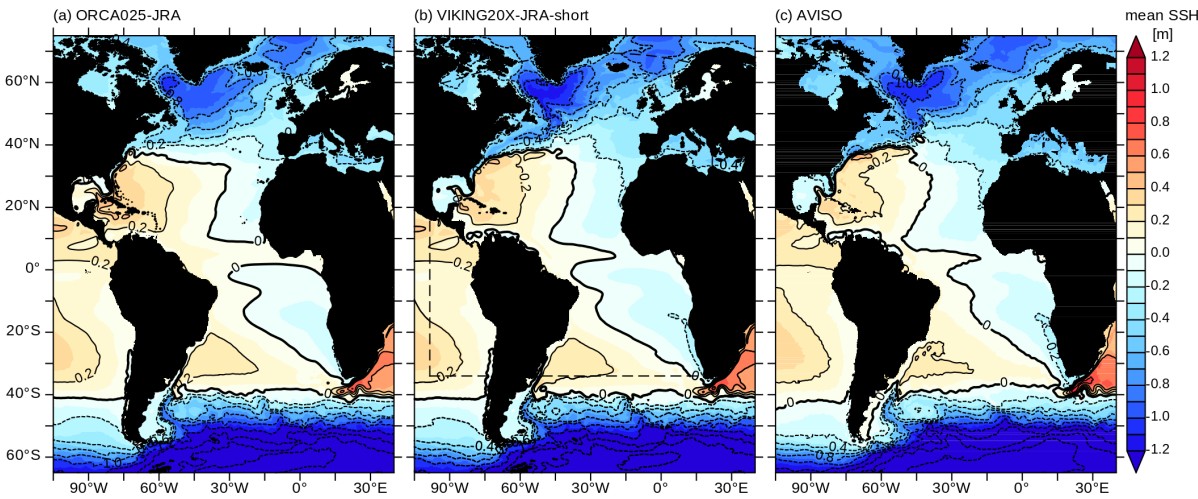

**Figure A1.** Mean (1993-2019) sea surface height in (a) ORCA025-JRA, (b) VIKING20X-JRA-short and (c) satellite altimetry.

*Author contributions.* AB and CB defined and guided the overall research problem and methodology; FUS and KG developed and performed the ocean model simulations. FUS, KG, SR, TM, MS, TS, PH and RH contributed analyses and figures for the individual parts. All coauthors discussed the analyses and contributed to the text.

*Competing interests.* The authors declare that they have no conflict of interest.

*Acknowledgements.* The ocean model simulation was performed at the North-German Supercomputing Alliance (HLRN) and on the Earth System Modelling Project (ESM) partition of the supercomputer JUWELS at the Jülich Supercomputing Centre (JSC). We thank the NEMO system team for support. This research has been supported by the German Federal Ministry of Education and Research (grants SPACES-CASISAC (03F0796A) and RACE-Regional Atlantic Circulation and Global Change (03F0729C)). It has also received funding from the Initiative and Networking Fund of the Helmholtz Association through the project 'Advanced Earth System Modelling Capacity (ESM)' and the European Union's Horizon 2020 research and innovation programmes under grant agreements No 818123 (iAtlantic) and No 817578 (TRIATLAS). The Florida Current cable and section data are made freely available on the Atlantic Oceanographic and Meteorological Labo-





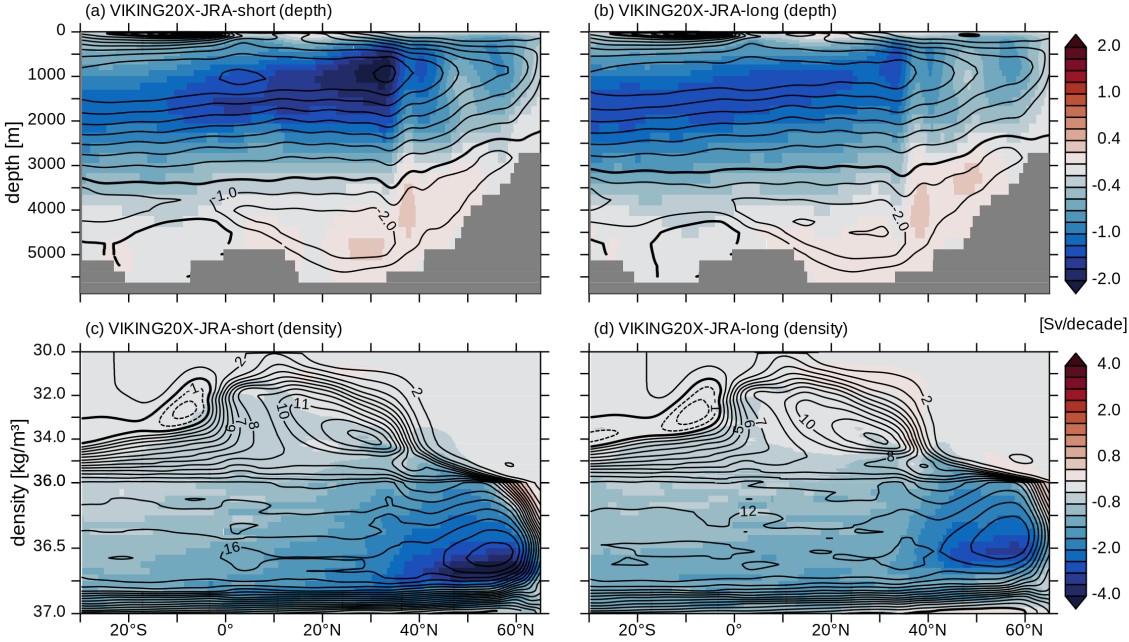

**Figure A2.** Mean (contours) and linear trend (shading) in the AMOC in (a and b) depth and (c and d) density ($\sigma_2$) coordinates for the period 2000 to 2019 for VIKING20X-JRA-short and VIKING20X-JRA-long. All fields are meridionally smoothed with a Hanning filter of 10° window size.

ratory web page (www.aoml.noaa.gov/phod/floridacurrent/) and are funded by the DOC-NOAA Climate Program Office - Ocean Observing and Monitoring Division.



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
