# Peer review of "Regional Imprints of Changes in the Atlantic Meridional Overturning Circulation in the Eddy-rich Ocean Model VIKING20X"

_Ocean Science, 2021_

## Referee Comment (RC2)

**Review of manuscript number os-2021-37**

Regional Imprints of Changes in the Atlantic Meridional Overturning Circulation in the Eddy-rich Ocean Model VIKING20X

by Arne Biastoch et al.

**General impression**
The paper discusses a suite of hindcast simulations at varying resolution and atmospheric forcing. It systematically describes the key features of the AMOC and the boundary currents contributing to it, and as such is a valuable benchmark for follow-up studies.

I only have a few minor comments on the text and figures (mostly requests for clarification) and a list of text suggestions / typos. I therefore recommend **minor revisions** before publication in Ocean Science.

**Minor comments**

**Writing style:**
The authors sometimes have a tendency to make long sentences with several clauses, combining a lot of information for a reader to digest in one go. Often, such sentences also contain information on more than one subject / topic. Splitting them is usually easy, and will certainly improve the readability of the paper. Some examples are listed below, but I urge the authors to review the entire paper with this in mind.

Examples: 17-20; l.31-35; l.49-52, l.96-102, l.166-168, l.221-223, l.515-519

**Detecting AMOC changes in regional current systems:**
Besides a systematic description of the various numerical simulations, the authors also present an analysis of the modelled trends in the AMOC and in its current components at various latitudes, to address at which latitudes one can expect to see changes if these occur. I like that aim and the outcomes, but I find it not carefully and clearly phrased in the manuscript. Places where this needs some attention:

Abstract l.9-10: *"Regional observations in western boundary current systems at 53N, 26.5N and 11S are explored in respect to their ability to represent the AMOC and to monitor the temporal evolution of the AMOC"*

This can be misread as id the authors are solely analyzing observations. In stead, using the outcomes of the various model simulations, they explore if observations at these locations have the ability to give us a good view on / quantification of potential  AMOC changes.

l.86-90: *A particular emphasis of the study is on the imprints of AMOC variability and trends on WBC systems which, in turn, contributes to exploring the capability of regional observation systems to capture changes in the basin-scale AMOC. Using the different evolution of the experiments in respect to the long-term evolution of the AMOC, we turnaround the question and ask which regional observations are able to capture changes in the AMOC.*

In the final sentence, it can be made more explicit that the authors analyze the features of the model simulations at the locations where long-term observations exist

L.621 *"our main objective is to test whether regional current systems are able to detect AMOC changes."*

I find this is oddly phrased, as the AMOC at a certain latitude is by definition (the net effect of) what happens in regional current systems. In my view, the key aspect addressed here is if an AMOC change would appear as a clear signal or not at these locations

**Questions**

l.260: I do not understand the statement that is made here: do the authors claim that the features of the horizontal circulation are less important for the AMOC than forcing / thermohaline drivers? And that this is because horizontal circulation = set by grid resolution? [which I would find a strange reasoning – one could argue that if it is key to get certain features horizontal circulation correctly represented then this implies using a certain minimal resolution is crucial]. Please elaborate / rephrase

L.409-413: The conclusion that the reduction in MLD_a has to be due to model drift confuses me, as convective activity in itself does not necessarily represent a changing contribution to the AMOC [in theory, as long as the dense water is not exported, the contribution of convection to the AMOC is zero].
Yet the authors apparently not only expect a one-to-one connection between convection and the AMOC strength, as they are suprised this is not happening (l.410) , but for these two specific experiments this one-to-one connection can also act in the reverse direction? (AMOC reduction yields convection reduction l. 412).
To me, this comes across as a too simplified view of a highly complex current system in which the AMOC, convection regions and the subpolar gyre interact, i.p. in eddy-rich models. Please elaborate / clarify

l.690
It probably helps that all currents are cyclonic at 53N so no compensation effects between upper and deep ocean that potentially confound signals?  (unlike f.ex 11 S – Fig 16)

**Figures**

- Fig A1: a difference plot would help to support the statement on l.251
- Fig 3 – why in black and white?  Hard to see the differences
- Fig 10 – consider adding the zero-line in the plot for clarity
- Fig 11 grey density lines hardly contrast with blue colors
- Fig 14 authors never refer to this figure
- Fig. 17d-e appear very cluttered, impossible to distinguish lines

**Text suggestions & typos**

- l.4-6: re-order sentence: The representation of the Atlantic Meridional Overturning Circulation (AMOC), and in particular its long-term temporal evolution, strongly depends on numerical choices for the application of freshwater fluxes.
- l.8 – pointing at a dominant role of the forcing – this is an important remark, which is now presented in passing. In my view, it could use a half sentence explanation
- l.11 if → provided
- l.22 I find this sentence oddly phrased (order of words is not grammatically correct, also why most difficult?). Please rewrite
- l.23 attempts aim -very vague
- l.26 This is an "apples & oranges" list: some items mention the current that is observed, some just the name / location of the array but not what is measured there. Please make consistent
- l.58 odd sentence, please rephrase [I think my confusion comes from the use of "integrated" here as "performing a numerical calculation" rather than as "unified"]
- l.63 not clear to me what "these" refers to in this sentence
- l.64 the use of "thermohaline events" is unclear to me – what is meant with that? Why not simply the thermohaline circulation?
- l.82 improve, compared to what? VIKING20X versus VIKING20? Or compared to older / lower resolution models?
- l.84-86 Not clear to me what is meant; what exactly is exploited / explored. Please rephrase
- l.86-88 sentence not clear / grammatically not correct: "WBC systems, which… contribute" [but is that what the authors want to say?]
- l.89 turn around / reverse
- l.93 results
- l.95 start the section with some guidance for the reader what is coming; after this intro text the section header of 2.1 seems incomplete / not covering the content
- l.98 features
- l.102 the successful representation of the physical circulation
- l.111 strangely phrased with 2x compare and 2x AMOC in the sentence
- l.124 remove "to"; with → at
- l.130 an → a
- l.134 damped by applying
- l.144 sentence unclear
- l.165 add comma after level
- l.188 represented differently on the two ocean model grids
- l.190 In section 2.2 acronyms are used distinguishing the various model experiments (VIKING20X-JRA-OMIP, ORCA025-JRA-OMIP, VIKING20X-JRA-short, VIKING20X-JRA-long) but the list of experiments is only given in section 2.3.
- Table 1 - not sure why the long names are given? Seems only relevant to the modellers?
- l.222 unclear, something is missing? – with a piston velocity of 50 m 4.1 yr$^1$
- l.223 – In…. suppressed: unclear to me what is meant here
- l.235 2x systematic in one sentence
- l.240 repeats l.216; it is unclear to me what point the authors want to make in l.240-243
- l.244/246 is focus is on basin-scale AMOC then title section 3 should reflect that
- l.251 explain what aspect is improved [basically the text on l.256-259, as this is about the mean and not about the variability / Fig 2]
- l.255 remove "at" or "around"
- l.265- are → is

- l. 267 re-order sentence for clarity: The strength of the NADW cell in ORCA025-JRA is quite different from that in VIKING20X-CORE, with 1990-2009 average values at 26.5N ranging from 10.9 Sv to 20.4, respectively (Table 2).
- Table 2 note on *: rephrase; values indicated by * denote a shorter averaging period … due to the limited length of the CORE forcing and RAPID data sets
- L.273 – does the statement belong here, if it is discussed later, and no reference is given to a figure supporting it? Or specify where "below" is?
- L.277 specify where "below" is
- L.293 add "in the various experiments"
- L.304 robust in the models and compares well to / in good agreement with observations?
- L.306 1980-2009
- L.309 variability of the AMOC strength
- L.309-311 connection between the different statements can be clarified / made more explicit [now it reads as a loose set of statements]. Guess the point is to discuss the differences in variability on various timescales between RAPID and the models. Also: what do the authors conclude from the fact that these differ?
- L.317 trend in …
- L.321 indicate how you define "Arctic FWC " – which latitudes; in text and caption of Fig 6; same for subpolar FWC – l.326
- L.322 the trend … is stable → that does not seem correct – looks like there is no trend; the FWC is stable
- L.322 … and increasing..: not sure one can say the trend in increasing based on just the figure, but the FWC itself certainly is. Please clarify what is meant
- L.325 maybe add the trends in km^3 /yr that emerge from the models for comparison
- l.326 as l.321 - indicate how you define "subpolar FWC " – which latitudes; in text + caption of Fig 6
- l.335 sentence unclear – needs a comma after 'correction' ?
- l.339 capable of
- l.345 indicate which experiment; indicate how this pathway is deduced/defined [referring to it as a pathway to me suggests something Lagrangian, which is not the case]
- l.346-7 oddly phrased - the path is broken into eddies… and reconfigures..
- l.355 remove brackets
- l.355 statement is a loose end, any attempt to explain?  Van Sebille uses Lagrangian techniques – can this be relevant?
- L.355 A concluding statement on section 3 is missing
- L.356 title not very informative – regional imprints of what?
- L.358-365 could use some references to review papers…
- L.366-370 missing discussion on densities in 1/20deg versus 1/4deg – linking this to the resolution / mixing argument presented earlier
- L.383-385: not clear what statement the authors want to make – please elaborate why is this is noteworthy
- L.386 "additional deepwater is added to the system" - please rephrase – additional to what? Which system?
- L.390 mention criterium used to define the bottom boundary of the MLD
- L.402 unclear which experiment 'former' refers to
- L.430 similar to above remark: "the AMOC… 1990s" – this seems to suggest the authors expect a one-to-one connection between the AMOC and gyre strength. Is that indeed the author's statement? If so then explain why
- L.486 are → is
- L.486 the availability of long-term observations for comparison?

- L. 488 modelled transports agree; but statement based on Table 6 seems at odds with curves shown in Fig 14
- L.505-this needs some more specific guidance for the reader what to look for; what exactly is "indicating" and "suggesting" this underlying physical explanation in terms of a connection to NBC rings
- L. 513 figure domain ends before that latitude
- L.527 provide overestimate in % is more common?
- L.534 last
- Fig 17 – The text only refers to Fig 17a – l.537, and to 17e (as a sidestep on l.534). Fig 17b-c-d are never mentioned
- L.556 this → these
- L.567 signal,
- L.562 indicate how one can see this from ssh (variance)
- L.566 19a
- L.580-582 I understand what the authors mean, but I find the sentence not very clear (probably because it starts with "depending on" after which I expect something like "you may either see A or B"
- L.587 "forcing comes an important change" - unclear what is meant here
- L.589 could → can?
- L.591 "as well as its regional components' – unclear what is meant here
- L.597 / 593 – not clear if the authors make the connection here between SPG transport and AMOC strength? Please clarify what point authors want to make on l.592-599
- L.599 ability to simulate
- L.601 features of dc regions
- L.630 and l. 635 explain how you know what the forcing mechanism is in this simulation (or is this an assumption? Then rephrase )
- L.634 We identify – explain how & where / indicate in plot
- L.638 only slightly slanted – they look quite straight to me – so why not an explanation in terms of surface forcing?
- l.639 Only or Besides
- l.640 give latitude
- l.654 in principle – I'd say it is really a matter of trend amplitude versus variability? Of does 'in principle' imply 'if the timeseries is sufficiently long' ?
- l.559- the transport … AMOC – something is missing in this sentence
- l.673 bracket missing
- l.674 "At 11S, the WBC system is more exposed to the open ocean": not sure what is meant here
- L.694 add reference to Moat et al 2020 - https://doi.org/10.5194/os-16-863-2020
- L.705 "An important factor could be the 5th cycling of the simulations through the forcing period" – please rephrase / explain what this means for readers not familiar with the procedure
- L.727 concentrate on
- Typos in references - author names l.824, 873, 965

---

## Author Comment (AC1)

**Author's response to Anonymous Referee #1**

*Original reviewer's comments are inserted in black, Author Replies (AR) are added in blue, and Changes made to the Manuscript (CM) are finally listed in grey, whereby line numbers refer to the fully revised version of the manuscript.*

**General Comments:**

The authors use the NEMO ocean model to carry out a set of simulations that have 1/4 degree horizontal resolution across the entire globe, as well as some which feature AGRIF nests to achieve 1/20 degree horizontal resolution within the Atlantic Ocean. These simulations are then compared against observations that span the North and South Atlantic Ocean. Observed AMOC transports at multiple locations are compared against the model output, identifying certain features. The authors find that surface freshwater restoring influences the AMOC transport. The manuscript is well thought out, covers a much larger expanse of the Atlantic than we normally read about in AMOC-related articles, and provides a lot of information that modellers, observationalists, and climate scientists may find useful.

AR: We thank you for your kind reply and constructive criticism below which helped to improve the manuscript.

**Specific Comments:**

- L135-137: Is this free/no slip change for both the parent and nest, or just the nest? If only the nest, I'm curious why no-slip isn't included everywhere in the nest as it helps generate eddies. Perhaps include some text in the manuscript on why only use a small region with no-slip.

  AR: One of the major achievements in NEMO was the implementation of an energy-enstrophy conserving momentum advection scheme in combination with partial cells. However, as shown for ORCA025 by Penduff et al. (2007), the use of no-slip sidewall boundary conditions cancelled most of the improvements. The following developments with ORCA025 and subsequent high-resolution nests like VIKING20X were usually continued with free-slip boundary conditions. During the development of VIKING20X, we performed sensitivity experiments which confirmed the better performance of, e.g. the circulation in the Northwest Corner, with free-slip conditions. However, it was also noticed that the generation of Irminger Rings was only possible through no-slip conditions in the region of the West Greenland Current (Rieck et al., 2019). This only applies to the nest. The motivation of these choices is now included in the text.

  CM (L.136-140): Horizontal sidewall boundary conditions are formulated as free-slip, which was demonstrated as an optimal choice in ORCA025 to benefit from the energy-enstrophy conserving momentum advection scheme in combination with partial cells (Penduff et al., 2007). Sensitivity tests during the development of VIKING20X confirmed that free-slip is also the preferred option at higher resolution. In contrast, the generation of West Greenland Current eddies were shown to work best with no-slip conditions (in the nest) which motivated its use in a region around Cape Desolation (Rieck et al., 2019).

- L150: I find it intriguing that the spatial multiplier is 5 between ORCA025 and VIKING20x, but you were able to get away with a time multiplier of 3. I always thought AGRIF required the same

spatial/temporal multiplier even though you set it. I might have to try this later. No response needed here, I just wanted to express I learned something.

AR: Technically, AGRIF does not require a fixed ratio of spatio and temporal refinements. The choice of the time step depends on fulfilling the CFL Criterion in the respective grid. For global grids the largest possible timestep is usually determined by the smallest (usually most poleward) grid cell. If a nested domain is not directly placed at high latitudes, the timestep can usually be chosen larger than the grid refinement would suggest.

- L229-236: It isn't clear what the INATL20-JRA-long simulation will provide. Assessing nested boundary condition issues is useful, but this simulation has different restoring, slip conditions, and tides. Please provide more justification here on why this simulation is included, particularly in reference to any anticipated AMOC changes suspected due to nested boundary conditions.

AR: We are aware that the INALT20-JRA-long simulation is not a systematic sensitivity experiment. Due to the fact that the southern boundary of the high-resolution nest in VIKING20X cuts right through e.g. the Cape Basin where most of the variability has an origin south and east of it, we wanted to explore the impact of the nest extension and the lack of properly represented eddies outside on the mesoscale and interannual variability in its direct vicinity. For both scales, the differences in the numerical setting are less important. In the new version, we have justified the inclusion of the INALT20-JRA-long simulation as follows:

CM (L.240-243): Owing to the differences in the numerical setting, we concentrate the comparison to the influence of the Agulhas Current system and the Malvinas confluence region on the mesoscale and interannual variability. Both are represented in VIKING20X only on the coarser host grid but are part of the nested high-resolution region in INALT20.

- Figure 9: The spatial area of the MLD is much different between these runs. But I don't see how their MLD volume (d) is so similar. This is addressed in L403-413, but I still have an issue with the MLD plots, I would have expected to see a larger change in fig 9d. Perhaps it is because the spatial max MLD is plotted. Perhaps try plotting a mean of the annual max MLD?

AR: The colour shading in Fig. 9a-c does indeed show long-term (1980-2009) mean of the annual maximum MLD. The MLD structure is quite similar in both ORCA025 and VIKING20X. Details in depth and horizontal pattern obviously compensate, so that, under the same forcing, it results in similar total volume. This is already mentioned in the text:

CM (L.416-419): While the resolution seems to determine the general spatial structure, the forcing and other model specific settings impact the intensity and temporal variability of deep convection. During the first 15 years, i.e., until the mid 1970s, the $MLD_a$ volume in the depicted domain (Fig. 9d) shows nearly the same magnitude and temporal variability for all simulations. (notably, the overall mixed layer volume in the ORCA025 simulations is not systematically larger than in the VIKING20X simulations).

- L355: Any suggestion why this pathway is not seen in your simulations?

AR: Thank you for pointing that out. Indeed, this was an inconsistent comparison between Lagrangian water mass spreading focusing on the path of the NADW towards the Indian Ocean (Van Sebille et al., 2012) and overall horizontal velocities (here). In an experiment using Lagrangian particles we can identify a similar path towards the Indian Ocean as in Van Sebille et al. (2012).

CM (L.359-363): A zonal band of slightly elevated speed between 20° S and 25° S indicates the flow of NADW water described by Van Sebille et al. (2012) which could also be identified in our simulations by Lagrangian experiments (here not shown).

**Technical corrections:**

- Figure 2c- My PDF viewer shows horizontal white lines on the AVISO figure, but they are only over land and do not make the figure difficult to view.

AR: Thank you for this hint. We updated Figure 2, eliminating the white lines.

CM: Figure 2 was updated.

- L21-22: Grammar issue clouds this sentence.

  AR: We agree and have changed the sentence to:

  CM (L.22-23): In contrast to its importance, the AMOC and its past evolution is most difficult to obtain and to quantify.

- L23: 'The RAPID array at 26.5N is ...'

  AR: Has been changed.

- L120: I'm not sure what an 'eddy-present' configuration is. Do you mean eddy-permitting or eddy-resolving? It isn't clear here or in the later parts of the manuscript

  AR: The terms "eddy-present" and "eddy-rich" follow recent nomenclature (e.g. Hewitt et al., 2020) replacing "eddy-permitting" and "eddy-resolving" by more unambiguous terms.

- L386-389: awkward sentence that could use a rewrite.

  AR: We agree and have rewritten accordingly:

  CM (L.400-402): In the subpolar North Atlantic, further deepwater is generated and added to the NADW. Owing to strong wintertime heat loss, in particular through strong and cold winds, the Labrador and Irminger Seas are regions of deepwater formation. Deep convection provides a lighter, upper component to the NADW (in contrast to the overflows forming lower NADW).

- L618: extra space before a period.

  AR: Has been changed.

- L639: 'Only Besides' is confusing, seems like the authors meant to write 'Besides...' and forgot to remove 'Only'.

  AR: That is correct, "Only" was removed

- L676: bit awkward "... does equal the one ..."

  AR: Has been changed:

  CM (L.693-694): for VIKING20X-JRA-long the NBUC trend is equal to the AMOC trend,

**References**

Hewitt, H. T., Roberts, M., Mathiot, P., Biastoch, A., Blockley, E., Chassignet, E. P., Fox-Kemper, B., Hyder, P., Marshall, D. P., Popova, E., Treguier, A. M., Zanna, L., Yool, A., Yu, Y., Beadling, R., Bell, M., Kuhlbrodt, T., Arsouze, T., Bellucci, A., Castruccio, F., Gan, B., Putrasahan, D., Roberts, C. D., Van Roekel, L., and Zhang, Q.: Resolving and Parameterising the Ocean Mesoscale in Earth System Models, Curr. Clim. Chang. Reports, 6, 137–152, https://doi.org/10.1007/s40641-020-00164-w, 2020.

Penduff, T., Le Sommer, J., Barnier, B., Treguier, A. M., Molines, J. M., and Madec, G.: Influence of numerical schemes on current-topography interactions in 1/4° global ocean simulations, Ocean Sci., 3, 509–524, https://doi.org/10.5194/os-3-509-2007, 2007.

Rieck, J. K., Böning, C. W., and Getzlaff, K.: The nature of eddy kinetic energy in the labrador sea: Different types of mesoscale eddies, their temporal variability, and impact on deep convection, J. Phys. Oceanogr., 49, 2075–2094, https://doi.org/10.1175/JPO-D-18-0243.1, URL http://journals.ametsoc.org/doi/10.1175/JPO-D-18-0243.1, 2019.

Van Sebille, E., Johns, W. E., and Beal, L. M.: Does the vorticity flux from Agulhas rings control the zonal pathway of NADW across the South Atlantic?, J. Geophys. Res. Ocean., 117, C05 037, https://doi.org/10.1029/2011JC007684, URL http://doi.wiley.com/10.1029/2011JC007684, 2012.

---

## Author Comment (AC2)

**Author's response to Anonymous Referee #2**

*Original reviewer's comments are inserted in black, Author Replies (AR) are added in blue, and Changes made to the Manuscript (CM) are finally listed in grey, whereby line numbers refer to the fully revised version of the manuscript.*

**General impression**

The paper discusses a suite of hindcast simulations at varying resolution and atmospheric forcing. It systematically describes the key features of the AMOC and the boundary currents contributing to it, and as such is a valuable benchmark for follow-up studies. I only have a few minor comments on the text and figures (mostly requests for clarification) and a list of text suggestions / typos. I therefore recommend minor revisions before publication in Ocean Science.

AR: We thank you for your general impression and the series of constructive comments which allowed us to improve the manuscript. We also thank you for the careful reading and the series of hints to improve the text.

**Minor comments**

- Writing style: The authors sometimes have a tendency to make long sentences with several clauses, combining a lot of information for a reader to digest in one go. Often, such sentences also contain information on more than one subject / topic. Splitting them is usually easy, and will certainly improve the readability of the paper. Some examples are listed below, but I urge the authors to review the entire paper with this in mind. Examples: 17-20; l.31-35; l.49-52, l.96-102, l.166-168, l.221-223, l.515-519

  AR: We carefully went through these suggestions and the whole manuscript and rewrote towards shorter sentences.

- Detecting AMOC changes in regional current systems: Besides a systematic description of the various numerical simulations, the authors also present an analysis of the modelled trends in the AMOC and in its current components at various latitudes, to address at which latitudes one can expect to see changes if these occur. I like that aim and the outcomes, but I find it not carefully and clearly phrased in the manuscript. Places where this needs some attention:

  Abstract l.9-10: "Regional observations in western boundary current systems at 53N, 26.5N and 11S are explored in respect to their ability to represent the AMOC and to monitor the temporal evolution of the AMOC" This can be misread as id the authors are solely analyzing observations. In stead, using the outcomes of the various model simulations, they explore if observations at these locations have the ability to give us a good view on / quantification of potential AMOC changes.

  AR: We agree and have clarified:

  CM (L.8-10): The ability of the model to represent regional observations in western boundary current systems at 53° N, 26.5° N and 11° S is explored. It is analysed if WBC systems are able to represent the AMOC and to monitor the temporal evolution of the AMOC.

  l.86-90: A particular emphasis of the study is on the imprints of AMOC variability and trends on WBC systems which, in turn, contributes to exploring the capability of regional observation systems to capture changes in the basin-scale AMOC. Using the different evolution of the experiments in

respect to the long-term evolution of the AMOC, we turnaround the question and ask which regional observations are able to capture changes in the AMOC.

In the final sentence, it can be made more explicit that the authors analyze the features of the model simulations at the locations where long-term observations exist

AR: This part now reads:

CM (L.87-89): A particular emphasis of the study is on the AMOC variability and trends and its imprint in WBC systems. Using the different evolution of the experiments in respect to the long-term evolution of the AMOC, we ask if the regional observations are able to capture changes in the AMOC.

L.621 "our main objective is to test whether regional current systems are able to detect AMOC changes."

I find this is oddly phrased, as the AMOC at a certain latitude is by definition (the net effect of) what happens in regional current systems. In my view, the key aspect addressed here is if an AMOC change would appear as a clear signal or not at these locations

AR: We agree and have clarified:

CM (L.636-637): Beyond the description and verification of the VIKING20X set of experiments, our main objective is to test whether regional current systems are able to detect latitudinally coherent AMOC changes.

**Questions**

- l.260: I do not understand the statement that is made here: do the authors claim that the features of the horizontal circulation are less important for the AMOC than forcing / thermohaline drivers? And that this is because horizontal circulation = set by grid resolution? [which I would find a strange reasoning – one could argue that if it is key to get certain features horizontal circulation correctly represented then this implies using a certain minimal resolution is crucial]. Please elaborate / rephrase

AR: Thanks for pointing this out, this was indeed unclear. We have reformulated:

CM (L.266-268: The horizontal circulation is largely determined by the grid resolution and the wind field. In contrast, the vertical overturning circulation does not only depend on the atmospheric forcing but also on details of its application such as SSS restoring and its impact on the freshwater budget at higher latitudes (Behrens et al., 2013).

- L.409-413: The conclusion that the reduction in $\mathrm{MLD}_a$ has to be due to model drift confuses me, as convective activity in itself does not necessarily represent a changing contribution to the AMOC [in theory, as long as the dense water is not exported, the contribution of convection to the AMOC is zero]. Yet the authors apparently not only expect a one-to-one connection between convection and the AMOC strength, as they are suprised this is not happening (l.410) , but for these two specific experiments this one-to-one connection can also act in the reverse direction? (AMOC reduction yields convection reduction l. 412). To me, this comes across as a too simplified view of a highly complex current system in which the AMOC, convection regions and the subpolar gyre interact, i.p. in eddy-rich models. Please elaborate / clarify

AR: We agree that a one-to-one relation between $\mathrm{MLD}_a$ and AMOC cannot be drawn, in particular without calculating deepwater formation and export. This is in particular true in the light of the recent findings by Lozier et al. (2019) who suggest a larger influence of the eastern subpolar North Atlantic to the AMOC than the Labrador Sea. A detailed analysis is beyond this study and underway. We have clarified the corresponding part.

CM (L.422-426): A causal relation between $\mathrm{MLD}_a$ and AMOC is subject to discussion in the light of the recent observational findings by Lozier et al. (2019) and subject to a separate study. Here we note that in VIKING20X-JRA-long and ORCA025-JRA the decrease of $\mathrm{MLD}_a$ volume sets in after the simulated AMOC decline described above. At least parts of the diagnosed negative $\mathrm{MLD}_a$ volume trends in VIKING20X-JRA-long and ORCA025-JRA can be attributed to spurious model drifts.

- l.690: It probably helps that all currents are cyclonic at 53N so no compensation effects between upper and deep ocean that potentially confound signals? (unlike f.ex 11 S – Fig 16)

  AR: That could be the case. However, in contrast to the discussion around 11S, we do not think that these thoughts would require an extensive analysis which is beyond the scope.

**Figures**

- Fig A1: a difference plot would help to support the statement on l.251

  AR: Similar to a streamfunction, a map of sea surface height mainly reflects the general structure of the horizontal circulation. A difference map (Fig. R1) does not help to explore the differences. We rather feel that the differences are better seen in the original figure and opt to not include a difference map.

[Figure]

Figure R1: Difference of sea surface height in VIKING20X-JRA-short (Fig. A1b) and ORCA025-JRA (Fig. A1a) with mean contours from AVISO (Fig. A1c)

- Fig 3 – why in black and white? Hard to see the differences

  AR: Black and white was chosen by intention to emphasise the fact that ocean transports are related to the gradients of the streamfunction, rather than the values itself. We also feel that the transition from the clock-wise NADW cell to the counter clock-wise AABW cell is better visible. The numbers of the strength are given in Table. 2

- Fig 10 – consider adding the zero-line in the plot for clarity

  AR: The zero line has been added.

- Fig 11 grey density lines hardly contrast with blue colors

  AR: The figure has been revised, the density lines read more clearly.

- Fig 14 authors never refer to this figure

  AR: Has now been referenced in lines 501 and 508.

- Fig. 17d-e appear very cluttered, impossible to distinguish lines

  AR: We have revised Fig. 17. Panels d and e have been expanded and should now be easier to read, see Figure R2

[Figure]

Figure R2: (Revised figure 17) Time series of interannually filtered (a) AMOC, (b) NBUC and (c) DWBC transports at 11° S; monthly averages of positive (NNE-ward) NBUC and negative (SSW-ward) DWBC transports (see Table 7 for details of the definitions) are given in (d) and (e) together with mooring based (orange curves; thick monthly, thin 2.5-day averages) and ship based observations (orange dots)

**Text suggestions & typos**

- l.4-6: re-order sentence: The representation of the Atlantic Meridional Overturning Circulation (AMOC), and in particular its long-term temporal evolution, strongly depends on numerical choices for the application of freshwater fluxes. AR: Has been done.

- l.8 – pointing at a dominant role of the forcing – this is an important remark, which is now presented in passing. In my view, it could use a half sentence explanation AR: We have made this a separate sentence.

- l.11 if → provided AR: Has been changed.

- l. 22 I find this sentence oddly phrased (order of words is not grammatically correct, also why most difficult?). Please rewrite AR: Has been done. CM (L.22-23): In contrast to its importance, the AMOC and its past evolution is most difficult to obtain and to quantify.

- l.23 attempts aim -very vague AR: has been changed. CM (L.24): Several observational projects monitor the AMOC at specific latitudes.

- l.26 This is an "apples & oranges" list: some items mention the current that is observed, some just the name / location of the array but not what is measured there. Please make consistent AR: The list now contains the names of the arrays.

- l.58 odd sentence, please rephrase [I think my confusion comes from the use of "integrated" here as "performing a numerical calculation" rather than as "unified"] AR: "integrated" has been replaced by "performed".

- l.63 not clear to me what "these" refers to in this sentence AR: Has been clarified.

- l.64 the use of "thermohaline events" is unclear to me – what is meant with that? Why not simply the thermohaline circulation? AR: Correct, has been changed.

- l.82 improve, compared to what? VIKING20X versus VIKING20? Or compared to older / lower resolution models? AR: Has been clarified: CM (L.83-84): ...a configuration that realistically simulates various key aspects of wind-driven and thermohaline ocean dynamics.

- l.84-86 Not clear to me what is meant; what exactly is exploited / explored. Please rephrase AR: Has been clarified: CM (L.85-87): We will explore the AMOC sensitivity on such parameters aided by a set of experiments differing in choices of the forcing (i.e., based on CORE and JRA55-do), initial conditions, and some aspects of the formulation of the freshwater fluxes.

- l.86-88 sentence not clear / grammatically not correct: "WBC systems, which... contribute" [but is that what the authors want to say?] AR: Has been clarified.

- l.89 turn around / reverse AR: Has been corrected.

- l.93 results AR:Has been corrected.

- l.95 start the section with some guidance for the reader what is coming; after this intro text the section header of 2.1 seems incomplete / not covering the content AR: An intro sentence has been added: CM (L.95-96): This study utilises output from the eddy-present global model ORCA025 and the eddy-rich nested configuration VIKING20X, both performed under atmospheric forcings of the past decades to simulate the Atlantic Ocean circulation.

- l.98 features AR: Has been corrected.

- l.102 the successful representation of the physical circulation AR: Has been corrected.

- l.111 strangely phrased with 2x compare and 2x AMOC in the sentence AR: Has been corrected.

- l.124 remove "to"; with → at AR: Has been corrected.

- l.130 an → a AR: Has been corrected.

- l.134 damped by applying AR: Has been corrected.

- l.144 sentence unclear AR: Has been clarified.

- l.165 add comma after level AR: Has been corrected.

- l.188 represented differently on the two ocean model grids AR: Has been corrected.

- l.190 In section 2.2 acronyms are used distinguishing the various model experiments (VIKING20X-JRA-OMIP, ORCA025-JRA-OMIP, VIKING20X-JRA-short, VIKING20X-JRA-long) but the list of experiments is only given in section 2.3. AR: A reference to next section and Table 1 has been included here.

- Table 1 - not sure why the long names are given? Seems only relevant to the modellers? AR: The long names are important for other (in particular external) scientists working with the model output. It allows them to directly refer the filenames to the experiments described here. The short names may change in other manuscripts, but the internal names are unique identifiers.

- l.222 unclear, something is missing? – with a piston velocity of 50 m 4.1 yr1 AR: The formulation of the piston velocity for a 50-m layer is a typical model standard, independent of the uppermost vertical grid cell. We have expanded the formulation.

- l.223 – In.... suppressed: unclear to me what is meant here AR: Has been replaced by "not applied".

- l.235 2x systematic in one sentence AR: Second "systematic" has been replaced by "consistent".

- l.240 repeats l.216; it is unclear to me what point the authors want to make in l.240-243 AR: We have removed the OMIP protocol in this second instance and have argued with the Tsujino et al. (2020) reference.

- l.244/246 is focus is on basin-scale AMOC then title section 3 should reflect that AR: We start this section with the horizontal circulation and would argue that both horizontal circulation and AMOC can be referred as "Basin-wide circulation".

- l.251 explain what aspect is improved [basically the text on l.256-259, as this is about the mean and not about the variability / Fig 2] AR: We see this sentence as some sort of introduction, in particular since the mean figure is in the supplementary material (but given for completeness). The differences are much better visible in the variability (Fig. 2), hence are described there.

- l.255 remove "at" or "around" AR: "at" removed

- l.265- are → is AR: Has been corrected.

- l. 267 re-order sentence for clarity: The strength of the NADW cell in ORCA025-JRA is quite different from that in VIKING20X-CORE, with 1990-2009 average values at 26.5N ranging from 10.9 Sv to 20.4, respectively (Table 2). AR: To underline the general resolution dependency, we have followed your suggestions, but used the general terms ORCA025 and VIKING20X instead.

- Table 2 note on *: rephrase; values indicated by * denote a shorter averaging period ... due to the limited length of the CORE forcing and RAPID data sets AR: Has been changed.

- L.273 – does the statement belong here, if it is discussed later, and no reference is given to a figure supporting it? Or specify where "below" is? Yes, we would like to already mention it at this stage. A reference to section 4.1 is given.

- L.277 specify where "below" is AR: A reference to section is given.

- L.293 add "in the various experiments" AR: has been added.

- L.304 robust in the models and compares well to / in good agreement with observations? AR: Has been added.

- L.306 1980-2009 AR: Has been corrected.

- L.309 variability of the AMOC strength AR: Has been added.

- L.309-311 connection between the different statements can be clarified / made more explicit [now it reads as a loose set of statements]. Guess the point is to discuss the differences in variability on various timescales between RAPID and the models. Also: what do the authors conclude from the fact that these differ? AR: We have expanded. It now reads: CM (L.315-317): However, the interannual variability of VIKING20X is higher than that of ORCA025, pointing to the importance of mesoscale variability for the interannual timescale. However, VIKING20X still underestimates the observations by more than 30%.

- L.317 trend in ... AR: "in AMOC strength" has been added.

- L.321 indicate how you define "Arctic FWC" – which latitudes; in text and caption of Fig 6; same for subpolar FWC – l.326 AR: We have revised the figure which now contains the region definitions

[Figure]

Figure R3: (Revised figure 6) Pentadally filtered (a) Arctic and (b) subpolar Freshwater Content (computed from seawater alone, hence excluding sea-ice and snow volume, using a reference salinity of $S_{ref}$ =34.7, in $10^3$ km$^3$). Note the inverted y-axis. Regions for the integration are shown as inlets.

- L.322 the trend ... is stable → that does not seem correct – looks like there is no trend; the FWC is stable AR: Correct, "The trend" has been removed.

- L.322 ... and increasing..: not sure one can say the trend in increasing based on just the figure, but the FWC itself certainly is. Please clarify what is meant AR: Correct, similar as above.

- L.325 maybe add the trends in km$^3$ /yr that emerge from the models for comparison AR: Linear trends (1992 to 2012) in Arctic freshwater content from JRA55-do forced experiments in VIKING20X are: VIKING20X-JRA-short: 819 km$^3$ /yr; VIKING20X-JRA-long: 985 km$^3$ /yr; VIKING20X-JRA-OMIP: 439 km$^3$ /yr. We have added the values to the text.

- l.326 as l.321 - indicate how you define "subpolar FWC" – which latitudes; in text + caption of Fig 6 AR: Done, see above.

- l.335 sentence unclear – needs a comma after 'correction' ? AR: has been added.

- l.339 capable of AR: Has been corrected.

- l.345 indicate which experiment; indicate how this pathway is deduced/defined [referring to it as a pathway to me suggests something Lagrangian, which is not the case] AR: We have clarified this. VIKING20X experiments are quite similar, so that a specific mentioning does not make sense in the text. Instead, it is given in the caption of Figure 7. CM (L.353): Figure 7 illustrates the circulation in NADW layer, indicating the spreading from the Nordic Sea...

- l.346-7 oddly phrased - the path is broken into eddies... and reconfigures.. AR: We have added a comma and changed to "reconfines".

- l.355 remove brackets AR: Has been corrected.

- l.355 statement is a loose end, any attempt to explain? Van Sebille uses Lagrangian techniques – can this be relevant? AR: Correct, this point was also raised by Reviewer 1. It now reads: CM (L.361-363): A zonal band of slightly elevated speed between $20°$ S and $25°$ S indicates the flow of NADW water described by Van Sebille et al. (2012) which could also be identified in our simulations by Lagrangian experiments (here not shown).

- L.355 A concluding statement on section 3 is missing AR: We agree and have added: CM (L.364-368): From the evaluation in this section, we conclude that the VIKING20X configuration successfully represents the basin-wide horizontal circulation. The strength of the overturning circulation is sensitive to the atmospheric forcing and numerical choices. VIKING20X-CORE, VIKING20X-JRA-OMIP and VIKING20X-JRA-short simulate a realistic AMOC strength, whereby only the latter two cover the recent years. In the following section we will explore how different AMOC evolutions are reflected in regional current systems.

- L.356 title not very informative – regional imprints of what? AR: "of AMOC Changes" have been added.

- L.358-365 could use some references to review papers... AR: Some key references (Schott and Brandt, 2007; Marshall and Schott, 1999; Böning et al., 2006; Dickson and Brown, 1994; Blindheim and Oster-hus, 2005) have been added.

- L.366-370 missing discussion on densities in 1/20deg versus 1/4deg – linking this to the resolution / mixing argument presented earlier AR: We think that a discussion at this stage would not help, in particular since the density differences arise further south as a consequence of downslope flow and subsequent entrainment. However, this is certainly worth to be mentioned. We therefore included for the discussion at 53N: CM (L.460-462): VIKING20X compares better to the observations than ORCA025, which is the effect of a combination of a denser overflow (Table 3) and a better representation of the downslope flow and its associated entrainment of ambient water masses.

- L.383-385: not clear what statement the authors want to make – please elaborate why is this is noteworthy AR: We have clarified this. It now reads:CM (L.397-399): It is interesting to note that the first cycle in ORCA025-JRA-OMIP shows a similar stabilisation in overflow density after about 25 years. The second cycle is subject to a lighter density which is also reflected in a weaker AMOC (Fig. 5). In contrast, VIKING20X-JRA-OMIP does not show such a spin-down in overflow density nor AMOC strength.

- L.386 "additional deepwater is added to the system" - please rephrase – additional to what? Which system? AR: We have rephrased. It now reads CM (L.400): In the subpolar North Atlantic, further deepwater is generated and added to the NADW.

- L.390 mention criterium used to define the bottom boundary of the MLD AR: The mixed layer depth is defined as sigma difference of 0.01 with respect to $10\,$m depth. The definition is given in the caption of Figure 9.

- L.402 unclear which experiment 'former' refers to AR: Has been clarified.

- L.430 similar to above remark: "the AMOC... 1990s" – this seems to suggest the authors expect a one-to-one connection between the AMOC and gyre strength. Is that indeed the author's statement? If so then explain why AR: We have omitted this part of the sentence.

- L.486 are → is AR: Has been corrected.

- L.486 the availability of long-term observations for comparison? AR: We think that this sentence reads correct.

- L. 488 modelled transports agree; but statement based on Table 6 seems at odds with curves shown in Fig 14 AR: We have clarified this. It now reads: CM (L.506-508): In contrast to the pure metric (Table 6), we find a wide range of temporal variability. Some of the JRA55-do experiments are correlated on interannual timescales, but none of the experiments is correlated with the observations (Fig. 14).

- L.505-this needs some more specific guidance for the reader what to look for; what exactly is "indicating" and "suggesting" this underlying physical explanation in terms of a connection to NBC rings AR: Has been clarified. A deeper understanding, however, is beyond the scope of this manuscript. Instead, we refer to the (now published) manuscript by Schulzki et al. (2021). CM (L.519-520): Between 10° N and 10° S, offshore recirculation patterns appear more prominently than at other latitudes which is indicated by broader WBC patterns.

- L. 513 figure domain ends before that latitude AR: This statement refers to Fig. 7 which included this latitude.

- L.527 provide overestimate in % is more common? AR: We prefer to keep the ratios (1/3 and 2/3).

- L.534 last AR: Has been corrected.

- Fig 17 – The text only refers to Fig 17a – l.537, and to 17e (as a sidestep on l.534). Fig 17b-c-d are never mentioned AR: References have been added. No additional text required.

- L.556 this → these AR: Has been corrected.

- L.567 signal, AR: Has been corrected.

- L.562 indicate how one can see this from ssh (variance) AR: The information was already there. But we have reformulated: CM (L.576-577): This is seen in more confined SSH pattern in Cape Basin (Fig. 18b and d) as an indicator of a too regular pathway into the South Atlantic.

- L.566 19a AR: Done

- L.580-582 I understand what the authors mean, but I find the sentence not very clear (probably because it starts with "depending on" after which I expect something like "you may either see A or B" AR: We agree and have reformulated accordingly. CM (L.594-596): Once the mesoscale is resolved by an adequate resolution over the full model domain and the model is driven by a realistic and balanced atmospheric forcing, many aspects of the simulated wind-driven and thermohaline circulation compare very well with observations.

- L.587 "forcing comes an important change" - unclear what is meant here AR: We have clarified: CM (L.600-601): The shift of the ocean model community from CORE forcing (Large and Yeager, 2009) to the higher resolved and updated JRA55-do forcing (Tsujino et al., 2018) is an important change.

- L.589 could → can? AR: Has been changed.

- L.591 "as well as its regional components' – unclear what is meant here AR: has been clarified: CM (L.605): ..., which is reflected in regional components.

- L.597 / 593 – not clear if the authors make the connection here between SPG transport and AMOC strength? Please clarify what point authors want to make on l.592-599 AR: We think that a short description about the robustness, but also about the limitations, of an eddy-present simulation is important. In addition to the mesoscale and iterannual variability, we now close with an additional sentence: CM (L.613-615): On decadal timescales differences between both resolutions play a more important role because of the better ability of VIKING20X to simulate key aspects of the thermohaline component such as overflow, entrainment and convection.

- L.599 ability to simulate AR: Has been corrected.

- L.601 features of dc regions AR: Has been added.

- L.630 and l. 635 explain how you know what the forcing mechanism is in this simulation (or is this an assumption? Then rephrase ) AR: Previous analyses (see references within the respective paragraph) have shown that interannual and decadal timescales can be referred to wind-driven and thermohaline components of the forcing, respectively.

- L.634 We identify – explain how & where / indicate in plot AR: We think that this is evident from Fig. 20.

- L.638 only slightly slanted – they look quite straight to me – so why not an explanation in terms of surface forcing? AR: Most of the lines are in fact slanted. However, this is discussed in detail in the lines before.

- l.639 Only or Besides AR: This was a typo, "Only" removed.

- l.640 give latitude AR: Has been given ($\sim$ 40-50$^\circ$ N)

- l.654 in principle – I'd say it is really a matter of trend amplitude versus variability? Of does 'in principle' imply 'if the timeseries is sufficiently long' ? AR: We have removed this half-sentence. The formulation is clear enough as it refers to decadal and longer-term AMOC trends.

- l.559- the transport ... AMOC – something is missing in this sentence AR: We have added a colon to make this more clear.

- l.673 bracket missing AR: Has been added.

- l.674 "At 11S, the WBC system is more exposed to the open ocean": not sure what is meant here AR: We have expanded and clarified this. It now reads: CM (L.691-692): At 11$^\circ$ S, AMOC changes cannot be indicated by changes in the WBC system alone, but also require the consideration of interior (Ekman) and eastern components (Rühs et al., 2015; Herrford et al., 2021).

- L.694 add reference to Moat et al 2020 - https://doi.org/10.5194/os-16-863-2020 AR: Has been added.

- L.705 "An important factor could be the 5th cycling of the simulations through the forcing period" – please rephrase / explain what this means for readers not familiar with the procedure AR: We have clarified this a few lines above: CM (L.717-718): Apart from the observation-model comparison, additional insight comes from the Ocean Model Intercomparison Project 2 (OMIP-2), in which models were performed six subsequent cycles under JRA55-do forcing.

- L.727 concentrate on AR: Has been corrected

- Typos in references - author names l.824, 873, 965 Have been corrected

**References**

[revised manuscript text omitted]